



# Potential Influences of Neglecting Aerosol Effects on the NCEP GFS Precipitation Forecast

Mengjiao Jiang[1,2], Jinqin Feng[3], Ruiyu Sun[4], Zhanqing Li[1,2*], Bingcheng Wan[5],

Maureen Cribb[2]

1, State Key Laboratory of Earth Surface Processes and Resource Ecology, College of

Global Change and Earth System Science, Beijing Normal University, Beijing, China

2, Department of Atmospheric and Oceanic Science and ESSIC, University of

Maryland, College Park, Maryland, USA

3, Longyan Meteorological Office of Fujian Province, Longyan, Fujian, China

4, IMSG, Environmental Modeling Center, National Centers for Environmental

Prediction, National Oceanic and Atmospheric Administration, USA

5, State Key Laboratory of Atmospheric Boundary Layer Physics and Atmospheric

Chemistry, Institute of Atmospheric Physics, Chinese Academy of Sciences, Beijing,

China

Correspondence to:

Z. Li,

zli@atmos.umd.edu



**Abstract**
Aerosol-cloud interactions (ACI) have been widely recognized as a factor affecting
precipitation. However, they have not been considered in the operational National
Centers for Environmental Predictions Global Forecast System model. We evaluated
the potential impact of neglecting ACI on the operational rainfall forecast using
ground-based and satellite observations, and model reanalysis. The Climate Prediction
Center unified gauge-based precipitation analysis and the Modern-Era Retrospective
analysis for Research and Applications, Version 2 aerosol reanalysis were used to
evaluate the forecast in three countries for the year 2015. The overestimation of light
rain (47.84%) and underestimation of heavier rain (31.83%, 52.94%, and 65.74% for
moderate rain, heavy rain, and very heavy rain, respectively) from the model are
qualitatively consistent with the potential errors arising from not accounting for ACI,
although other factors cannot be totally ruled out. The standard deviation of the
forecast bias is significantly correlated with aerosol optical depth in Australia, the
U.S., and China. To gain further insight, we chose the province of Fujian in China to
pursue a more insightful investigation using a suite of variables from gauge-based
observations of precipitation, visibility, water vapor, convective available potential
energy (CAPE), and satellite datasets. Similar forecast biases were found:
over-forecasted light rain and under-forecasted heavy rain. Long-term analyses reveal
an increasing trend of heavy rain in summer, and a decreasing trend of light rain in
other seasons, accompanied by a decreasing trend in visibility, no trend in water vapor,





and a slight increasing trend in summertime CAPE. More aerosols decreased cloud
effective radii for cases where the liquid water path was greater than 100 g m$^{-2}$. All
findings are consistent with the effects of ACI, i.e., where aerosols inhibit the
development of shallow liquid clouds and invigorate warm-base mixed-phase clouds
(especially in summertime), which in turn affects precipitation. While we cannot
establish rigorous causal relations based on the analyses presented in this study, the
significant rainfall forecast bias seen in operational weather forecast model
simulations warrants consideration in future model improvements.



## 1. Introduction

Aerosols affect precipitation by acting as cloud condensation nuclei (CCN) and ice nuclei (IN), which can influence cloud microphysics (Twomey et al., 1984) and cloud lifetime (Albrecht, 1989). By absorbing and scattering radiation in the atmosphere, aerosols can alter the thermal and dynamic conditions of the atmosphere. The two types of effects are broadly referred to as aerosol-cloud interactions (ACI) and aerosol-radiation interactions (ARI) (Intergovernmental Panel on Climate Change, 2013). Both can influence precipitation (Rosenfeld et al., 2008) and many other meteorological variables to the extent that they may account for the considerable changes in climate experienced in Asia over the past half century (Li et al., 2016).

The impact of aerosols on precipitation via cloud microphysics occurs through warm-rain and cold-rain processes, as reviewed by Tao et al. (2012). In the warm-rain process, the competition for water vapor leads to a greater number of cloud drops with smaller sizes as the aerosol loading increases. This decreases the collision efficiency because of the low fall speed and low droplet-collecting efficiency. Rain formation is thus slowed down. In addition, a heavier aerosol loading narrows the cloud drop-size spectrum, lowering the coalescence and collision efficiencies. In the cold-rain process, the delay in precipitation formation from the warm-rain process enhances condensation and freezing, and ultimately, leads to the release of extra latent heat above the $0\,^{\circ}\mathrm{C}$ isotherm (Andreae et al., 2004; Rosenfeld et al., 2008), favoring mixed-phase and cold rainfall processes. ARI also affect precipitation. First, solar radiation absorbed by aerosols may warm up a cloud droplet enough to evaporate it





(Ackerman et al., 2000). Second, heating of an aerosol layer due to absorption and
cooling of the surface because of the reduction in radiation reaching the ground
stabilizes the lower boundary-layer atmosphere and suppresses the formation and
development of low clouds whose occurrence decreases with increasing aerosol
loading (Li et al., 2011). The combination of ARI and ACI leads to a non-monotonic
response of rainfall to aerosols: increasing first and then decreasing (Jiang et al., 2016)
because the ACI and ARI are most significant for low and high aerosol loadings,
respectively (Rosenfeld et al., 2008; Koren et al., 2008; Fan et al., 2016).

Most findings concerning the aerosol suppression of clouds and precipitation are

associated with stratocumulus clouds, cumulus clouds, and shallow convection
(Albrecht, 1989; Rosenfeld 2000; Jiang et al., 2006; Xue and Feingold, 2006; Khain
et al., 2008), whereas those of enhanced rainfall are from deep convective clouds
(Koren et al., 2005; Lin et al., 2006; Bell et al., 2008; Rosenfeld et al., 2008). Li et al.
(2011) used 10 years of ground-based observations to examine the long-term impact
of aerosols on precipitation and found rainfall enhancement in mixed-phase
warm-base clouds and suppression in liquid clouds. Van den Heever et al. (2011)
underlined the importance of cloud type in dealing with the impact of aerosols on
precipitation.

Forecasting rainfall is most challenging and important in numerical weather

prediction (NWP). In the current Global Forecast System (GFS) model, aerosols are
only considered in the radiation scheme on a climatological scale. ARI are only
considered offline and are not coupled with the dynamic system. ACI have not been



accounted. To improve the forecast accuracy, a suite of new physical schemes are
being implemented in the National Centers for Environmental Prediction (NCEP)'s
Next-Generation Global Prediction System (NGGPS). The goal of modifying the
current forecast model is to improve physical parameterizations in such a way that
allows for efficient, accurate, and more complete representations of physical
processes and their interactions including at least some of the aforementioned
mechanisms.
As a first step, the goal of the present study is to evaluate current operational
GFS forecast results (before any ACI are introduced) to see if any systematic
precipitation biases bear resemblance to aerosol perturbations. A gross evaluation of
the GFS model forecast results in three countries (China, the U.S, and Australia) were
chosen, for they cover eastern and western hemispheres, northern and southern
hemispheres, and represent highly different atmospheric and environmental
conditions. Moreover, there are ARM observations in all three countries which will be
used in follow-on studies to gain a deeper insight of causal relationships and the
impact of different parameterization schemes. Descriptions of the operational GFS
model, datasets, and the evaluation strategy and statistical method used are presented
in section 2. Results of the evaluation and possible explanations are given in section 3.
A summary of the research and discussion are given in Section 4.

**2. Model, Datasets, and Methodology**
**2.1 Description of the NCEP GFS Model**





### 2.1.1 Model Basics


The NCEP GFS model is a global spectral (spherical harmonic basis functions)
model. The horizontal resolution is spectral triangular 1534 (T1534), or
approximately 13 km at the equator for days 0–10, and spectral triangular 574 (T574),
or approximately 34 km at the equator for days 10–16. The vertical domain is divided
into 64 sigma-pressure hybrid (Sela, 2009) layers with enhanced resolution near the
bottom and top (the top centered at about 0.27 hPa). The GFS model is based on the
primitive equations, which include vorticity and divergence equations, the mass
continuity equation, the hydrostatic equation, the thermodynamic equation, and the
water vapor equation with parameterizations for atmospheric physics (Kanamitsu,
1989; Yang et al., 2006). A prognostic cloud water scheme (Sundqvist et al., 1989;
Zhao and Carr, 1997; Moorthi et al., 2001) was added in May 2001.

### 2.1.2 Radiation


Shortwave and longwave radiation are parameterized using the Rapid Radiative
Transfer Models (RRTMG) RRTMG_SW (v2.3) and RRTMG_LW (v2.3),
respectively, developed at AER Inc. (http://www.emc.ncep.noaa.gov/GFS/doc.php). A
Monte Carlo independent column approximation method is used in the RRTMG to
deal with multi-layered clouds and a maximum-random cloud overlapping method is
assumed for radiative calculations (http://www.emc.ncep.noaa.gov/GFS/doc.php)
whose soundness has been assessed (Yoo et al., 2013). The cloud cover calculation for
radiation, which follows Xu and Randall (1996), was also modified because it





produced too much low cloud globally (Yoo et al., 2012, 2013). A monthly
climatology of aerosols composed of five primary species similar to that in the
Goddard Chemistry Aerosol Radiation and Transport model (GOCART) was used.
One or two major components were chosen for both longwave and shortwave
radiative transfer calculations.

**2.1.3 Planetary Boundary Layer**
In the planetary boundary layer (PBL), a hybrid eddy-diffusivity mass flux PBL
parameterization (Han et al., 2016) was incorporated to replace the previous PBL
scheme, which was originally proposed by Troen and Mahrt (1986)
(http://www.emc.ncep.noaa.gov/GFS/doc.php) and implemented by Hong and Pan
(1996). The PBL scheme was modified to improve daytime PBL growth
(http://www.emc.ncep.noaa.gov/GFS/doc.php).

**2.1.4 Convection**
A modified version (Han and Pan, 2011) of the Simplified Arakawa-Schubert
scheme (Arakawa and Schubert, 1974; Grell, 1993; Pan and Wu, 1995) is used for
deep convection in the GFS model. Water substance (liquid) detrained from the cloud
top is a source term of the prognostic cloud mixing ratio. The new shallow convection
scheme (Han and Pan, 2011) uses a bulk mass-flux parameterization, which is similar
to the deep convection scheme, but with a cloud-top limit of 700 hPa and different
specifications on entrainment, detrainment, and mass flux at the cloud base. The



detrained liquid water in updrafts is allowed to become convective rain (although the
precipitation from shallow convection is small) and grid-scale cloud condensate (Han
and Pan, 2011).

**2.1.5 Precipitation**

The cloud condensate has two sources: large-scale condensate (based on Zhao

and Carr (1997)), and convective condensation, which is from convective detrainment.
Convective precipitation is calculated from convection. Grid-scale precipitation is the
sink of cloud condensate and is diagnostically calculated from cloud condensate. It is
parameterized following Zhao and Carr (1997) for ice (snow), evaporation of rain and
snow, and the melting of snow, and following Sundvist et al. (1989) for liquid water
(rain) (GCWM Branch, EMC, 2003).

**2.2 Descriptions of Datasets Used**

Datasets used include Modern-Era Retrospective analysis for Research and

Applications, Version 2 (MERRA-2) aerosol optical depth (AOD) data, Climate
Prediction Center (CPC) unified gauge-based precipitation data, and the NCEP GFS
precipitation forecast data for the year 2015 in three countries: China, the U.S., and
Australia. Other datasets used include long-term NCEP Global Ensemble Forecast
System (GEFS) precipitation forecast data, ground-based observations of precipitation
and visibility, water vapor and convective available potential energy (CAPE)
sounding datasets, and satellite-retrieved aerosol and cloud properties for a small



region of Fujian Province in China chosen for more detailed study.

**2.2.1 NASA MERRA-2 Aerosol Reanalysis**

The MERRA-2 is the second generation of the MERRA reanalysis (Rienecker et

al., 2011). The biggest differences between the first and second versions of MERRA is
that the new generation of MERRA uses an updated model (Molod et al., 2012, 2015)
and a global statistical interpolation analysis scheme (Wu et al., 2002). This enables
the system to include new data types. MERRA-2 takes account of analyzed and
modeled aerosol fields with radiative effects that respond to the meteorological field
(Randles et al., 2016). The MERRA-2 aerosol reanalysis is an upgrade of the off-line
aerosol reanalysis called MERRAero (da Silva et al., 2011; Jiang et al., 2016). The
aerosol module in MERRAero is based on the GOCART model (Chin et al., 2002).
The bias-corrected AOD is retrieved from Moderate Resolution Imaging
Spectroradiometer (MODIS) observations. Cloud-screened AERONET AOD data are
used in the neural network to integrate MODIS radiances into bias-corrected AODs.
The MERRA-2 aerosol reanalysis includes additional measurements from the NASA
Earth Observing System, NOAA Polar Operational Environmental Satellites, and
ground-based observations (Randles et al., 2016). Bias-corrected AODs from satellite
and Aerosol Robotic Network (AERONET) AODs have been added in the
assimilation (Randles et al., 2016).The AOD observing system sensors extend from
the MODIS Neural Net Retrieval (NNR) in MERRAero to a combination of the
Advanced Very-High-Resolution Radiometer NNR, AERONET, the Multi-angle





Imaging SpectroRadiometer, the MODIS/Terra NNR, and the MODIS/Aqua NNR in
the MERRA-2 aerosol reanalysis. More details about the MERRA-2 aerosol
reanalysis can be found in Randles et al. (2016). Hourly total aerosol extinction AOD
data at 550 nm for the year 2015 are used in this study.

**2.2.2 CPC Unified Gauge-based Analysis of Global Daily Precipitation**
A unified suite of precipitation analysis products were assembled at NOAA's
CPC that ingest a gauge-based analysis of global daily precipitation over land
(https://climatedataguide.ucar.edu/climate-data/cpc-unified-gauge-based-analysis-glob
al-daily-precipitation). Over 30,000 station reports were first collected from multiple
sources. Quality control was performed through comparisons with other sources of
data, e.g., from radar, satellite, numerical models, independent nearby stations, and
historical precipitation records. Post-quality control corrected reports are interpolated
to create the analyzed fields. Orographic effects are considered in this step (Xie et al.,
2007). Finally, the daily analysis is constructed and released at a $0.5^{\circ}$ x $0.5^{\circ}$ resolution
(https://climatedataguide.ucar.edu/climate-data/cpc-unified-gauge-based-analysis-glob
al-daily-precipitation). Daily precipitation data for the year 2015 are used in this
study.

**2.2.3 NCEP GFS/GEFS Forecast Datasets**
The NWP model forecast data employed are three-hourly rainfall forecasts from
the NCEP GFS model initialized at 0000 coordinated universal time (UTC) and





accumulated for 24 hours in the three countries chosen for study. The mean cloud
mixing ratio at 850 hPa corresponding to the precipitation record in the U.S. at a
0.5°x0.5° latitude-longitude resolution for the year 2015 is also used in the analysis.
For the part of the study focused on Fujian Province, China, the NWP model
reforecast precipitation amount accumulated over the period of 12 hours to 36 hours
out from the 0000 UTC run at six-hourly intervals at a 1°x1° latitude-longitude
resolution for the years 1985 to 2010 are used to calculate the modeled daily
precipitation amount in each grid box. They are interpolated to match with long-term
ground-based precipitation observations recorded at each of the 67 stations in the
study region of Fujian, China (Fig. 1).

**2.2.4 Long-term Ground-based Observations in Fujian Province, China**

Ground meteorological data acquired in Fujian Province from 1980 to 2009 are

used in this study. Figure 1 shows the locations of the 67 meteorological stations
measuring precipitation. Sixteen of these stations also collect visibility data four times
a day. Daily mean data are employed here. Serving as a proxy for aerosol loading,
visibility was corrected for relative humidity (RH) (Charlson, 1969; Appel et al., 1985)
using the formula adopted by Rosenfeld et al. (2007) when RH falls between 40% and

99%:

$$\frac{V_{ori}}{V_{cor}} = 0.26 + 0.4285\, lg(100 - RH), \qquad (1)$$
where $RH$ is in percent, and $V_{ori}\ and\ V_{cor}$ are the originally uncorrected and
corrected visibilities, respectively. Only non-rainy data were used.





To analyze water vapor and atmospheric stability effects on precipitation, data
from three atmospheric sounding stations (Xiamen, 24.48ºN, 118.08ºE; Shaowu,
27.33°N, 117.46°E; Fuzhou, 26.08ºN, 119.28ºE) are used to calculate trends in
precipitable water vapor and CAPE. Daily precipitable water and CAPE values are
the mean of the two measurements made per day.

**2.2.5 Satellite Datasets of Aerosol and Cloud Properties in Fujian Province,**
**China**
CloudSat data from 2006–2010 amassed over Fujian Province (22.5ºN-28.5ºN,
114.5ºE-120.5ºE) are used to extract cloud-top and cloud-base height information.
CloudSat retrievals of cloud-top and base heights are converted to temperatures using
temperature profiles from the European Center for Medium-range Weather
Forecasting Auxiliary product. The converted cloud-top and cloud-base temperatures
are used for cloud type classification. The classification of different cloud types is
summarized in Table 1 and introduced in sub-section 2.3.1. Only single-layer clouds
detected by the CloudSat are chosen here.
Aqua/MODIS retrievals of cloud droplet size and liquid water path (LWP) for
liquid clouds (clouds with cloud-top temperatures (CTT) greater than 273 K) from
2003–2012 collected over Fujian Province are used. The MODIS Level 3 AOD at 550
nm product is also used. Grid boxes with AOD > 0.6 are excluded in this study to
reduce the possibility of cloud contamination in AOD retrievals.



### 2.3 Methodology

### 2.3.1 Rainfall Level Classification and Cloud Type Classification

Based on the definitions of the China Meteorological Administration, precipitation data are classified into four groups according to the daily rain amount: light rain (0.1–9.9 mm d$^{-1}$), moderate rain (10–24.9 mm d$^{-1}$), heavy rain (25–49.9 mm d$^{-1}$), and very heavy rain ($\geqslant$ 50 mm d$^{-1}$).

Table 1 summarizes the cloud types considered in the long-term analysis for Fujian Province. Deep mixed-phase clouds are defined as clouds with cloud-base temperatures (CBT) > 15°C and cloud-top temperatures (CTT) < -4°C, shallow mixed-phase clouds are defined as clouds with CBT ranging from 0°C to 15°C and CTT < -4°C, and pure liquid clouds are defined as clouds with CBT > 0°C and CTT > 0°C (Li et al., 2011; Niu and Li, 2012).

### 2.3.2 Evaluation Methods

Quantitative precipitation forecast scores developed by NCEP are used in the evaluation. Table 2 is a contingency table based on documents from the World Climate Research Programme (http://www.cawcr.gov.au/projects/verification/#Methods_for_dichotomous_forecasts). The most commonly-used statistical scores are the equitable threat score (ETS), which is also called the Gilbert skill score, and the bias score (BIAS). The ETS is given by

$$ETS = \frac{H - H_{random}}{H + m + f - H_{random}}, \tag{2}$$





where $H$ represents hits, $f$ represents false alarms, and $m$ represents misses. $H_{random}$
is given by

$$H_{random} = \frac{(H+m)*(H+f)}{TOTAL}. \qquad (3)$$

Its values range from -1/3 to 1 and a perfect score is 1. The BIAS is expressed as

$$BIAS = \frac{H+f}{H+m}. \qquad (4)$$

Its values range from 0 to infinity. A perfect score is 1. A BIAS $<$ 1 indicates
under-forecasting and a BIAS $>$ 1 indicates over-forecasting.
To obtain the forecast skill under a particular pollution condition, the ETS and
the BIAS for each AOD range are calculated as

$$< ETS >_{i,j,m} = (ETS)_{i,j,m}, \qquad (5)$$

$$< BIAS >_{i,j,m} = (BIAS)_{i,j,m}, \qquad (6)$$

for the index of precipitation threshold $i$, cloud mixing ratio $j$, and AOD bin $m$.

**2.3.3 Statistical Method**

The standard deviation of the precipitation bias between the GFS model and CPC
gauge data is calculated as

$$S = \sqrt{\frac{\sum(x-r)^2}{n-1}} \ , \qquad (7)$$

where $x$ is the forecast bias on a single day, $n$ is equal to 364 days, and $r$ is the mean
forecast bias. The Pearson's method is used to calculate the linear correlation
coefficient of the relationship between the standard deviation of the forecast
difference and AOD. A t-test is applied with the $p$ value set to 0.05.
The relative difference between the forecast precipitation and observations is





calculated as
$$\Delta P = \frac{P_{GFS/GEFS} - P_{OBV}}{P_{OBV}} \times 100\%, \qquad (8)$$
where $P_{GFS/GEFS}$ refers to the forecast precipitation and $P_{OBV}$ refers to the
precipitation from gauge-based observations.
For the long-term analysis, trends in a particular parameter are defined as the
relative change in the parameter (in %) over each successive decade (Lin and Zhao,
2009). The Mann-Kendall method is used to test the significance of the trend.

**3. Results**
**3.1 Evaluation of GFS Precipitation using the CPC Gauge-based Analysis**
**3.1.1 Annual Mean Patterns**
The CPC gauge-based precipitation analysis from 2015 is used to evaluate the
GFS precipitation forecast. Figure 2 shows the annual mean precipitation difference
between the GFS model and the CPC analysis for three countries, i.e., China, the U.S.,
and Australia, for the year 2015. Values above (below) zero represent the
overestimation (underestimation) of precipitation. In China (Figure 2a), the GFS
model overestimates the mean daily rainfall mostly in southwest China, especially in
Sichuan, Yunnan, and Guizhou Provinces (by ~3 mm d$^{-1}$), and in northwest China,
where rain events are scarcer. Rainfall is underestimated over the Yangtze River Delta
region and the eastern coast of China. In the U.S. (Figure 2b), the GFS model
overestimates precipitation by about 1–2 mm d$^{-1}$ in most regions and underestimates
precipitation along the coastline of the Gulf of Mexico (by ~1 mm d$^{-1}$). In Australia



(Figure 2c), the forecast performance is good. In northern Australia, the
underestimation of precipitation is around 2 mm d$^{-1}$. Z-scores were calculated to test
the significance of the annual mean difference in the daily rainfall amount between
the GFS model forecast and the CPC analysis. Z values range from -0.4803 to 0.8534
over the grids in the three countries. Because the Z-score values are less than 2, this
indicates that the mean difference is not significant at the two-sigma level. Therefore,
the forecast performance of the GFS model with regard to the annual mean daily
rainfall in the three countries is sound with reference to the gauge-based CPC rainfall
analysis.

**3.1.2 Different Rainfall Intensities**

Figure 3 shows the annual mean relative difference between forecast
precipitation and observations for light rain (0–10 mm d$^{-1}$) and heavier rain (> 10 mm
d$^{-1}$). The GFS model overestimates light rain in most places (Figure 3a) and
underestimates heavier rain (Figure 3b). This suggests that both the overestimation of
light rain and underestimation of moderate rain, heavy rain and very heavy rain
contribute to the forecast bias. Figure 4 shows the mean relative difference between
forecast and observed daily precipitation amounts for different rain intensities in the
three countries for whole year (Fig. 4a) and for summer only (Fig. 4b). GFS forecasts
overestimate light rain by 47.84% and underestimate moderate rain, heavy rain, and
very heavy rain by 31.83%, 52.94%, and 65.74%, respectively (Fig. 4a). The
underestimation of precipitation in summer is larger for moderate rain (32.93%),



heavy rain (55.19%), and very heavy rain (66.93%, Fig. 4b). Of course, these model
biases are caused by many factors, and it's beyond the scope of this paper to explore
all possible causes. Our focus is on any potential contribution by neglecting aerosol
effects to the biases. The relationship between model performance and AOD is thus
further investigated.

**3.1.3 Relationship between Model Performance and AOD**
In principle, the underestimation and overestimation at different rainfall levels
(Figs. 3 and 4) may be linked to AOD conditions, as elaborated in the introduction of
previous studies (c.f. the review of Tao et al., 2012). The standard deviation of the
forecast bias at each grid point in the three countries is calculated to further examine
the links between the model bias and AOD,as aerosols tend to polarize precipitation
by suppressing light rain and enhancing heavy rain and thus increase the standard
deviation. The calculation of the standard deviation of the forecast difference is based
on Eqn. (7). Figure 5 shows the relationship between the standard deviation and AOD
in the three countries. Each point represents a grid box. The standard deviation and
AOD has a significant positive correlation in the three countries with correlation
coefficients of 0.5602, 0.6522, and 0.5182 for Australia, the U.S., and China,
respectively. This suggests that the degree of disparity of the forecast error is larger
for regions with high aerosol loading. The slopes of the best-fit lines are 75.23 for
relatively clean Australia (maximum AOD < 0.18), 48.4 for the polluted U.S.
(maximum AOD < 0.20), and 8.554 for heavily polluted China (maximum AOD >



0.60).

The ETS and BIAS are used to examine the model performance in different
AOD bins for certain cloud mixing ratio conditions in the U.S. (Fig. 6). In Figs. 6a
and 6b, when the threshold is set to 5 mm d$^{-1}$, the ETS increases as the cloud mixing
ratio increases. This happens because large-scale precipitation is diagnostically
calculated from cloud mixing ratios. The ETS decreases as AOD increases except
under low cloud mixing ratio conditions. However, the BIAS shows little change as
AOD or the cloud mixing ratio changes. In Figs. 6c and 6d, when the threshold is set
to 20 mm d$^{-1}$, the ETS also increases as cloud mixing ratio increases. The ETS
decreases as AOD increases under all cloud mixing ratio conditions. This suggests
that the AOD influences the model rainfall forecast especially for stronger levels of
precipitation. The decreases in BIAS score with AOD (Fig. 6d) also shows that the
underestimation for heavy rainfall increases as AOD increases for low and middle
cloud mixing ratio conditions.

**3.2 Potential Contribution of Aerosols to the Model Bias**
**3.2.1 Long-term Forecast Bias and Trends in Observed Precipitation in Fujian**
**Province, China**
The model performance differs under different conditions, e.g., initial and
dynamic settings, and weather regimes. A long-term statistical evaluation of rainfall
forecasts for Fujian Province is made to mitigate these fluctuations in the model
forecast accuracy. Model data from 1985 to 2010 are used to calculate the relative



difference based on Eqn. (8). Figure 7 shows the mean relative difference between
forecast and observed precipitation for different rain rates from the 67 stations in
Fujian Province for all seasons and for summer only. Figure 7a shows that there is
114.36% more precipitation forecast by the NCEP/GEFS model than observed for the
light rain cases. For moderate rain, heavy rain, and very heavy rain cases, 29.20%,
41.74%, and 59.30% less precipitation than observed, respectively, was forecasted.
The underestimation of moderate rain (46.88%), heavy rain (59.58%), and very heavy
rain (70.16%) is even larger in summer (Fig. 7b).

Seasonally-averaged trends (percent change per decade) in daily rain amount and

frequency over Fujian Province from 1980 to 2009 are calculated. Only the results for
rain amount are shown in Fig. 8 because the frequency results bear a close
resemblance. Cross-hatched bars represent data at a confidence level greater than 95%.
In spring, daily rain amounts decreased over time, ranging from -4.9% to -15.3% per
decade for different rain rates. In summer, heavy and very heavy daily rain amounts
increased significantly. For very heavy rain, the amount and frequencies increased at a
rate of 21.8% and 24.5% (not shown), respectively. In autumn, light rain and
moderate rain amounts decreased. In winter, the light rain amount decreased over time.
Decreases in light rain amounts are -8.4% per decade. Overall, the increasing trends in
summertime for heavy and very heavy rain are most significant. The decreasing
trends in light rain in other seasons are also significant.

**3.2.2 Examination of Potential Contributors**



Reasons for the difference between modeled and observed precipitation are

examined in terms of aerosol effects, water vapor, and CAPE. The time series of

visibility over the period of 1980–2009 are shown in Fig. 9. Visibility has declined

steadily in all seasons but summer during which there was a short-lived increasing

trend from 1992–1997. The linear declining trends are statistically significant at the

95% confidence level. The greatest reduction is seen during the summer, especially

after 1997. Tables 3 and 4 summarize the correlation between visibility and

precipitation amount and frequency, respectively. A positive (negative) correlation

between visibility and precipitation means a negative (positive) correlation between

aerosol concentration and precipitation. Values with an asterisk represent data at a

confidence level greater than 95%. For light rain, the correlations between daily rain

amount and visibility (Table 3) and between rain frequency and visibility (Table 4) are

positive for all seasons. For heavy rain to very heavy rain, the correlations between

visibility and daily rain amount (Table 3), as well as frequency (Table 4), are negative

in summer.

Water vapor amount and atmospheric stability are important factors related to

precipitation. To analyze the potential contributions of these factors to the forecast

bias, their effects on precipitation are examined. Data from three atmospheric

sounding stations (Xiamen, 24.48ºN, 118.08ºE; Shaowu, 27.33°N, 117.46°E; Fuzhou,

26.08ºN, 119.28ºE) collected from 1980–2009 are used to calculate trends in

precipitable water vapor and CAPE. Figure 10 shows time series of annual mean

water vapor amount for different seasons. A slight increasing trend is seen in winter,



while no discernible trend is seen in other seasons. This suggests that the water vapor
amount characterizing the study region cannot explain seasonal variations in
precipitation. Time series of mean CAPE for the different seasons are shown in Fig.
11. There is an increasing trend in summertime CAPE during the period of 1980–2009,
but the trends are not as strong in other seasons. The observed increase in rain amount
in summer is in part likely due to an increase in convective precipitation events that
arises from the increasing trend in CAPE.

**3.2.3 Impact of Aerosols on Clouds and Precipitation**
Aerosols can influence precipitation through warm- and cold-rain processes (Tao
et al., 2012). Cloud droplet size, LWP for clouds with CTT greater than 273 K, and
AOD at 550 nm retrieved from the Aqua/MODIS platform over Fujian Province
during the period of 2003–2012 are used to examine the impact of aerosol on cloud
effective radius (CER). Figure 12 shows CER as a function of AOD for liquid clouds
with different LWPs. When the AOD is small (< 0.2), the CER increases with
increasing LWP. For LWP > 100 g m$^{-2}$, the CER decreases with increasing AOD,
which suggests that more aerosols decrease CERs. This result is in line with the two
aerosol indirect effects (Twomey et al., 1984; Albrecht, 1989). A greater number of
smaller droplets may reduce precipitation efficiency and suppress or enhance
precipitation, as reviewed by Tao et al. (2012).
Several observational and model studies suggest that smaller cloud particles are
more likely to ascend to above the freezing level, releasing latent heat and



invigorating deep convection (Rosenfeld et al., 2008; Li et al., 2011) while
suppressing shallow convection. Cloud top temperature (CTT) and cloud base
temperature (CBT), converted from CloudSat measurements of cloud top and base
heights, in Fujian Province from 2006 to 2010 are used to study the impact of aerosols
on the cloud development of different clouds. Figure 13 shows CTT as a function of
AOD for liquid and warm- and cold-base mixed-phase clouds. Definitions of the
different cloud types are summarized in Table 1, which is taken from Li et al. (2011).
Left-hand ordinates are for liquid clouds, while right-hand ordinates are for
warm-base and cold-base mixed-phase clouds. For all seasons (Fig. 13a), CTTs of
warm-base mixed-phase clouds are lower than those of cold-base mixed-phase clouds.
Warm-base mixed-phase CTTs decrease with increasing AOD, which indicates that
cloud-top heights have increased. For cold-base mixed-phase clouds, variations in
CTT with AOD are not obvious. For liquid clouds, CTTs increase slightly with AOD,
which means that the development of liquid clouds is suppressed when AOD
increases. In summer, CTTs decrease more significantly with increasing AOD for
warm-base mixed-phase clouds and increase more significantly with increasing AOD
for liquid clouds (Fig. 13b). This suggests that aerosols inhibit the development of
shallow liquid clouds and invigorate warm-base mixed-phase clouds, with little
influence on cold-base mixed-phase clouds. These effects of aerosols on summertime
cloud development are more obvious, likely because convective clouds occur more
frequently during the summertime in Fujian Province.
These results agree with those from a ground-based study using Atmospheric



Radiation Measurement Southern Great Plains data (Li et al., 2011) and from a
tropical region study using CloudSat/Cloud-Aerosol Lidar and Infrared Pathfinder
Satellite Observation data (Niu and Li, 2012; Peng et al. 2016). The impact of
aerosols on different types of clouds may lead to light rain suppression and heavier
rain enhancement. If the model neglects aerosol effects, the forecast may result in
overestimation for light rain and underestimation for heavy to very heavy rain. For
example, Fig. 14 shows time series of regionally-averaged daily modeled and
observed precipitation in 2001. Modeled and observed precipitation amounts over the
region agree well in spring and winter while modeled precipitation amounts are
greater than observations for light rain in autumn. Note that modeled precipitation
amounts are significantly less than observed precipitation amounts over the region in
summer when deep convective clouds and heavy to very heavy rain most likely occur.
Although there are many reasons for the difference between modeled and observed
precipitation, these results suggest that the neglect of aerosol effects may contribute to
the model rainfall forecast bias to some extent.

**Summary and Discussion**
Aerosol-cloud interactions (ACI) have been recognized as playing a vital role in
precipitation, but have not been considered in the National Centers for Environmental
Prediction (NCEP) Global Forecast System model yet. For more efficient and
accurate forecasts, new physical schemes are being incorporated into the NCEP's
Next-Generation Global Prediction System. As a benchmark evaluation of model



results that exclude aerosol effects, the operational precipitation forecast (before any
ACI are included) is evaluated using multiple datasets with the goal of determining if
there is any link between the model forecast bias and aerosol loading. Multiple
datasets are employed, including ground-based precipitation and visibility datasets,
Aqua/Moderate Resolution Imaging Spectroradiometer products, CloudSat retrievals
of cloud-base and cloud-top heights, Modern-Era Retrospective analysis for Research
and Applications, Version 2 model simulations of aerosol optical depth (AOD), and
GFS forecast datasets.

Operational daily precipitation forecasts for the year 2015 in three countries, i.e.,

Australia, the U.S., and China, were evaluated. The model overestimates light rain,
and underestimates moderate rain, heavy rain, and very heavy rain. The
underestimation of precipitation in summer is even larger. This is consistent
qualitatively with expected results because the model does not account for aerosol
effects on precipitation, i.e., the inhibition of light rain and enhancement of heavy rain
by aerosols. The standard deviations of forecast differences are generally positively
correlated with increasing aerosol loadings in the three countries. Equitable threat
scores also decrease with increasing AOD, especially for heavier rain forecasts.

An analysis of long-term measurements from Fujian Province, China was done.

Light rain overestimation, and moderate, heavy, and very heavy rain underestimations
from the Global Ensemble Forecast System were also seen. The underestimation for
stronger rainfall is larger in the summertime. Increasing trends for heavy and very
heavy rain in summer, and decreasing trends for light rainfall in other seasons are



significant from 1980 to 2009. Long-term analyses show that neither water vapor nor
convective available potential energy can explain these trends. Satellite datasets
amassed in Fujian Province from 2006 to 2010 were used to shed more light on the
impact of aerosols on cloud and precipitation. As implied by the Twomey effect, cloud
effective radius decreases with increasing AOD, which likely suppress light rain and
enhance heavy rain. Both of them can contribute to some extent to the model forecast
bias. The underestimation of heavy rain in summer most likely occurs because deep
convective clouds occur more frequently during the summertime.
How neglecting ACI in the operational forecast model impacts model biases
remains an open question. This study is arguably the first attempt at evaluating
numerical weather prediction forecast errors in terms of the potential effects of
aerosols. A more rigorous and systematic evaluation would require further insights
into the model with rich instantaneous measurements to allow for case-based
investigations that are under way.

**Data Availability**
Forecast data are from the NOAA NOMADS (https://nomads.ncdc.noaa.gov/)
for GFS data (https://nomads.ncdc.noaa.gov/data/gfs4/) and the NOAA NCDC
(https://www.ncdc.noaa.gov/data-access/model-data/model-datasets/global-ensemble-
forecast-system-gefs) for GEFS reforecast data. NASA MERRA-2 aerosol data are
accessible from the NASA Global Modeling and Assimilation Office
(https://gmao.gsfc.nasa.gov/reanalysis/MERRA-2/data_access/). The CPC Unified



Gauge-Based Analysis of Global Daily Precipitation dataset is available at
(https://climatedataguide.ucar.edu/climate-data/cpc-unified-gauge-based-analysis-glob
al-daily-precipitation). ECMWF reanalysis data are accessible via
http://apps.ecmwf.int/datasets/data/interim-full-daily/. MODIS data and CloudSat data
are available at https://modis.gsfc.nasa.gov/data/ and
http://www.cloudsat.cira.colostate.edu/, respectively. Ground-based observations of
precipitation amount, visibility, precipitable water, and CAPE from Fujian Province
can be requested from the Chinese Meteorological Administration's National
Meteorological Information Center (http://cdc.cmic.cn and http://data.cma.cn/).


**Acknowledgements**

This study was supported by the Ministry of Science and Technology of China

(2013CB955804), State Key Laboratory of Earth Surface Processes and Resource
Ecology (2015-TDZD-090), and NOAA (NA15NWS4680011). We would like to
thank the NASA Global Modeling and Assimilation Office
(https://gmao.gsfc.nasa.gov/reanalysis/MERRA-2/data_access/) and the Goddard
Space Flight Center Distributed Active Archive Center for their help in accessing
MERRA-2 inst3_2d_gas_Nx: 2d, 3-Hourly, Instantaneous, Single-Level, Assimilation,
Aerosol Optical Depth Analysis Version 5.12.4 data. We would also like to thank the
staff at the National Center for Atmospheric Research responsible for creating the
"The Climate Data Guide: CPC Unified Gauge-Based Analysis of Global Daily
Precipitation"
(https://climatedataguide.ucar.edu/climate-data/cpc-unified-gauge-based-analysis-glob





al-daily-precipitation). Thanks also go to the NOAA NOMADS
(https://nomads.ncdc.noaa.gov/) for GFS data
(https://nomads.ncdc.noaa.gov/data/gfs4/), the NOAA NCDC
(https://www.ncdc.noaa.gov/data-access/model-data/model-datasets/global-ensemble-
forecast-system-gefs) for GEFS reforecast data, and the NWS CPC for data
downloading software (http://www.cpc.ncep.noaa.gov/products/wesley/get_gfs.html).
We acknowledge the Chinese Meteorological Administration's National
Meteorological Information Center (http://cdc.cmic.cn and http://data.cma.cn/), the
European Centre for Medium-Range Weather Forecasts (ECMWF)
(http://www.ecmwf.int/), the NASA Goddard Space Flight Center
(https://modis.gsfc.nasa.gov/data/), and the CloudSat Data Processing Center
(http://www.cloudsat.cira.colostate.edu/) for providing the various datasets used in the
study.
We would also like to thank Drs. Yu-Tai Hou, Shrinivas Moorthi, and Jun Wang
from NOAA, Sarah Lu from State University of New York, Albany, Dr. Seoung-Soo
Lee and Lei Zhang from the University of Maryland, and Dr. Duoying Ji from Beijing
Normal University for their discussions regarding this study. We especially appreciate
the help given by Drs. Yu-tai Hou, Jongil Han, and Yuejian Zhu in understanding the
GFS/GEFS models and data products, and the guidance provided by Dr. Hye-Lim Yoo.
We also greatly appreciate the valuable comments from the anonymous reviewers.

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




## Figures and Tables


**Table 1.** Definitions of warm- and cold-base mixed-phase clouds and liquid clouds.

|  | Cloud-base temperature (ºC) | Cloud-top temperature (ºC) |
|---|---|---|
| Deep mixed-phase clouds with warm bases | > 15 | < -4 |
| Shallow mixed-phase clouds with cold bases | 0–15 | < -4 |
| Liquid clouds | > 0 | > 0 |



**Table 2.** Contingency table.

| Observed / Forecast | Observed yes | Observed no |
|---|---|---|
| Forecast yes | Hits | False alarms |
| Forecast no | Misses | Correct negatives |



**Table 3.** Correlation coefficients from linear regressions of visibility and different rain

amount types for all seasons.

| Season | Light rain | Moderate rain | Heavy rain | Very heavy rain | Rain amount |
|---|---|---|---|---|---|
| Spring | 0.48* | 0.51* | 0.48* | 0.17 | 0.40* |
| Summer | 0.08 | -0.16 | -0.28 | -0.41* | -0.38* |
| Autumn | 0.31 | 0.18 | 0.26 | -0.22 | 0.11 |
| Winter | 0.55* | 0.26 | 0.26 | 0.27 | 0.29 |

* Values with an asterisk represent data at a confidence level greater than 95%.






**Table 4.** Correlation coefficients from linear regressions of visibility and different
occurrence frequencies of rain amount type for all seasons.

| Rain rate / Season | Light rain | Moderate rain | Heavy rain | Very heavy rain | Rain amount |
|---|---|---|---|---|---|
| Spring | 0.61* | 0.51* | 0.38* | 0.08 | 0.67* |
| Summer | 0.23 | -0.13 | -0.26 | -0.44* | -0.04 |
| Autumn | 0.52* | 0.18 | 0.25 | -0.10 | 0.45* |
| Winter | 0.55* | 0.22 | 0.20 | -0.05 | 0.49* |

* Values with an asterisk represent data at a confidence level greater than 95%.












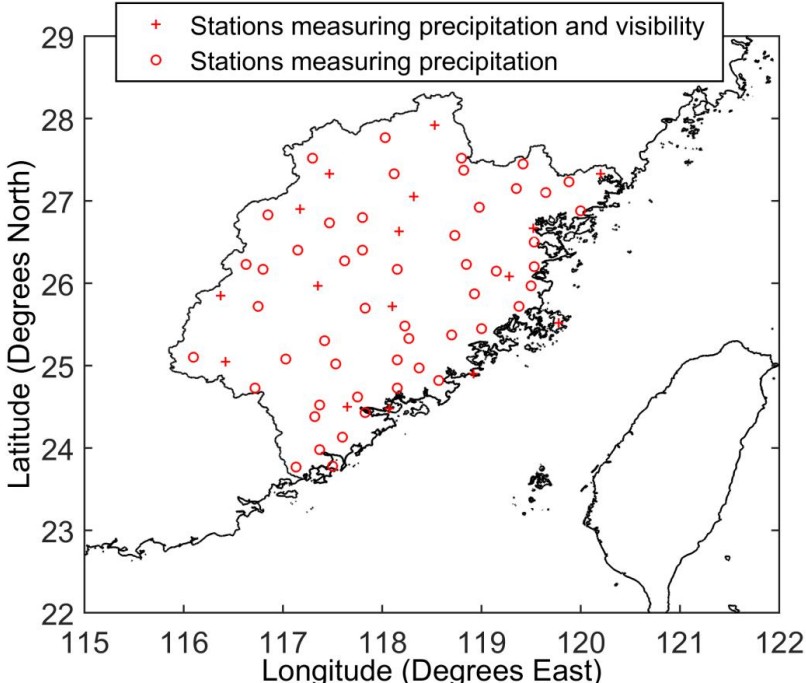


**Fig. 1.** Locations of 67 stations measuring precipitation in Fujian Province. Plus
symbols show the locations of the 16 stations where visibility measurements are also
made. This figure was plotted using the equidistant cylindrical projection.



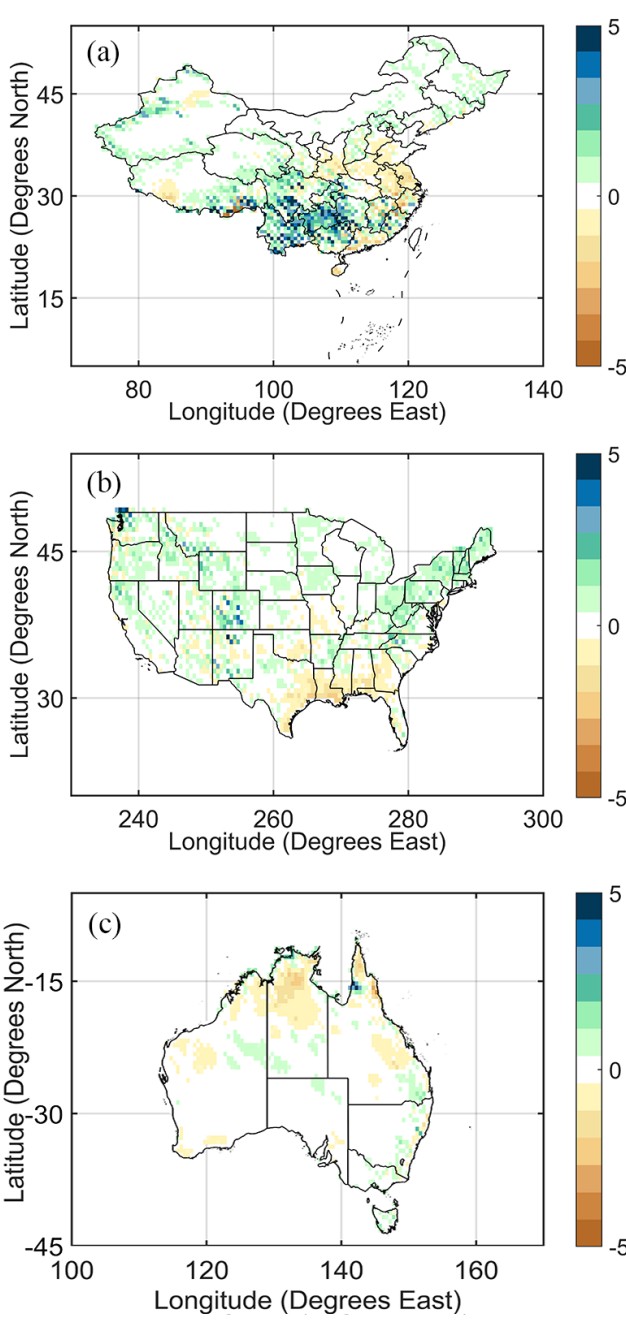

**Fig. 2.** Annual mean precipitation differences (in mm d$^{-1}$) between the GFS model
forecast and the CPC analysis in three countries: (a) China, (b) the contiguous U.S.,
and (c) Australia. Data are from the year 2015. This figure was plotted using the
equidistant cylindrical projection.





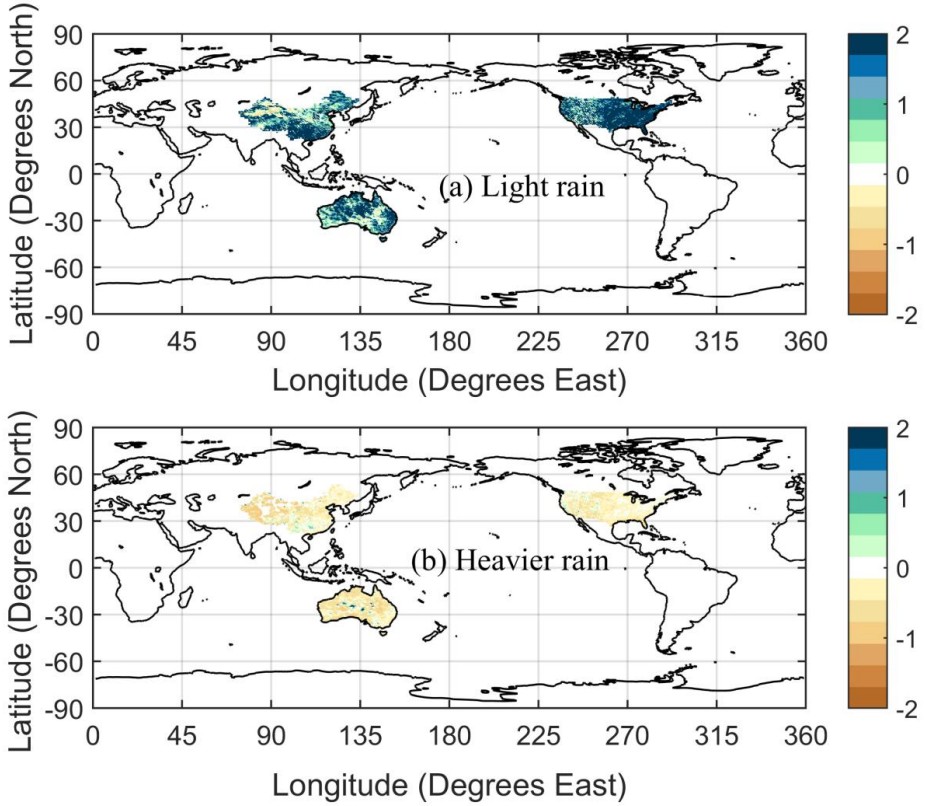


**Fig. 3.** Annual mean relative difference (in mm d$^{-1}$) between forecast and observed
precipitation for (a) light rain (< 10 mm d$^{-1}$) and (b) heavier rain (> 10 mm d$^{-1}$). Data
are from the year 2015. This figure was plotted using the equidistant cylindrical
projection.

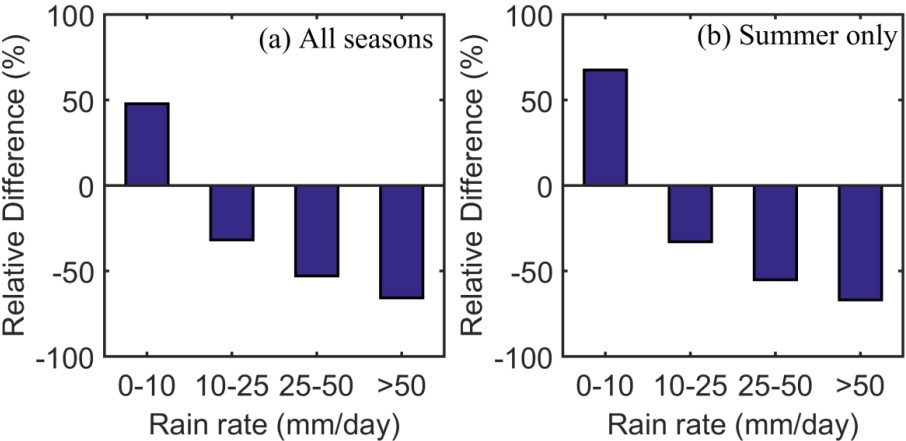

**Fig. 4.** Mean relative difference in precipitation between forecast and observed daily
light (< 10 mm d$^{-1}$), moderate (10–25 mm d$^{-1}$), heavy (25–50 mm d$^{-1}$), and very heavy
(> 50 mm d$^{-1}$) rain amounts for (a) all seasons and (b) summer only. Data are from the
year 2015 and from the three countries considered in the study.





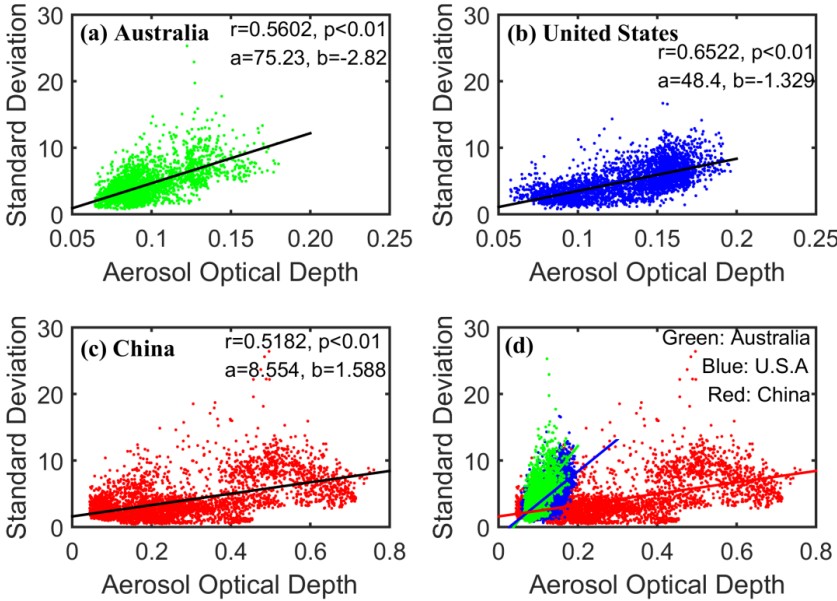


**Fig. 5.** Standard deviations of the daily precipitation difference as a function of
aerosol optical depth for (a) Australia (green points), (b) the United States (blue
points), (c) China (red points), and (d) all three countries. Data are from the year
2015.








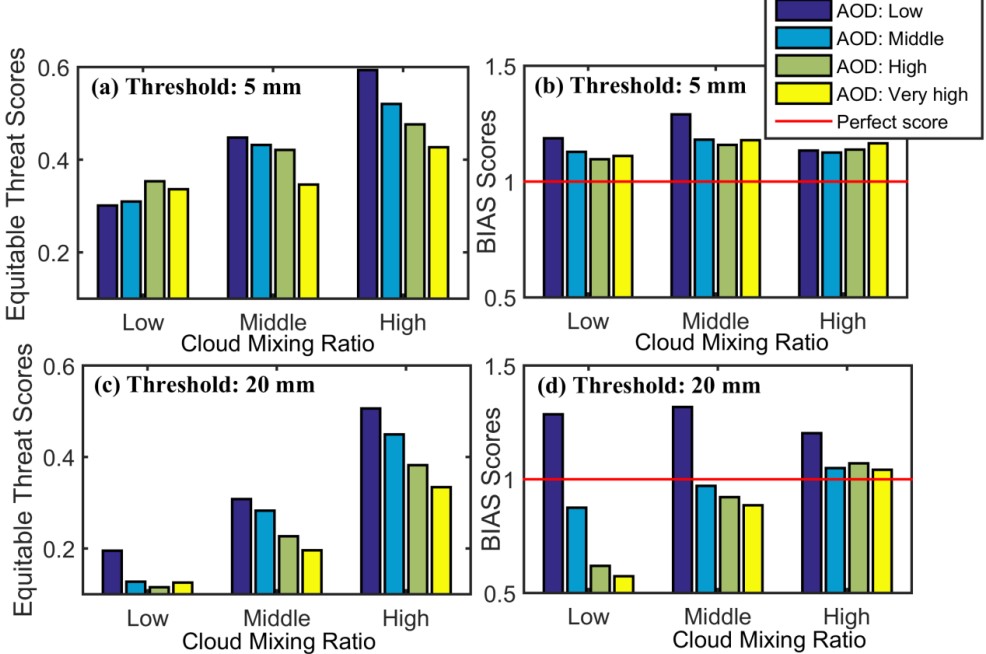


**Fig. 6.** ETS scores (a, c) and BIAS scores (c, d) in different AOD bins for certain cloud mixing ratio conditions. AOD is equally divided into four bins (low: dark blue bars; middle: blue bars; high: green bars; and very high: yellow bars). Cloud mixing ratios are equally divided into three categories (low, middle, and high). Data are from the year 2015 in the U.S. The horizontal red lines in (b) and (d) represent perfect scores.






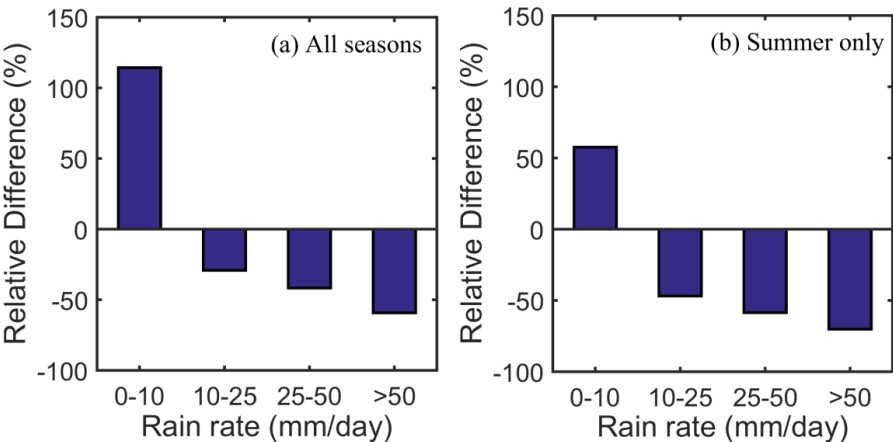


**Fig. 7.** Mean relative precipitation differences between forecast and observed daily light (< 10 mm d$^{-1}$), moderate (10–25 mm d$^{-1}$), heavy (25–50 mm d$^{-1}$), and very heavy (> 50 mm d$^{-1}$) rain amounts for (a) all seasons and (b) summer only in Fujian Province, China. Data are from 1985–2010.







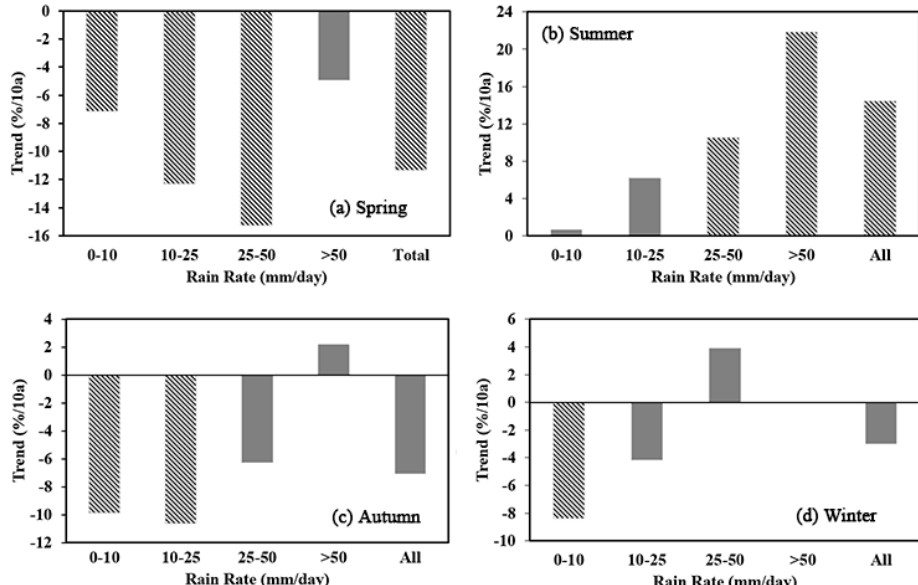


**Fig. 8.** Trends (percent change per decade) in mean daily light rain ($< 10$ mm d$^{-1}$),
moderate rain (10–25 mm d$^{-1}$), heavy rain (25–50 mm d$^{-1}$), very heavy rain ($> 50$ mm
d$^{-1}$), and total rain amounts for (a) spring, (b) summer, (c) autumn, and (d) winter in
Fujian Province, China. Data are from 1980–2009. Cross-hatched bars represent data
at a confidence level greater than 95%.





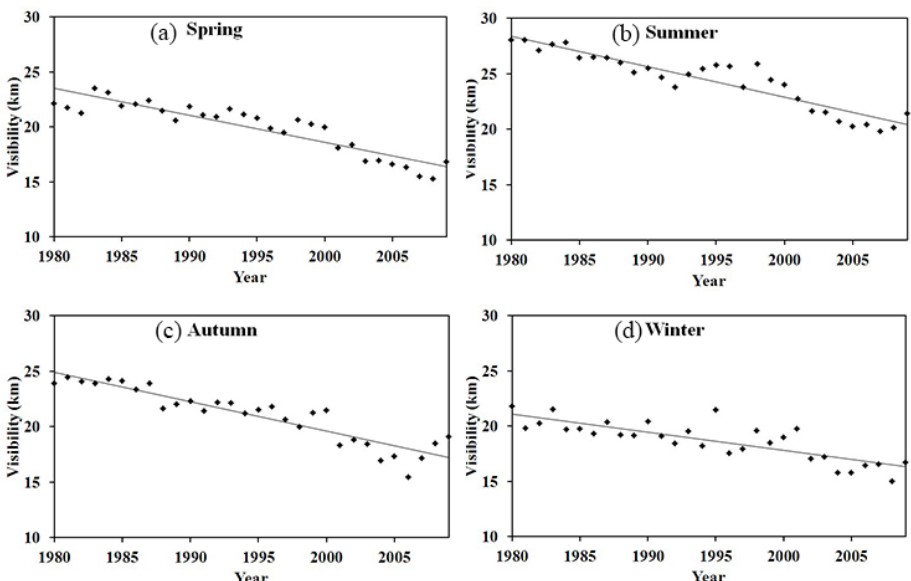


**Fig. 9.** Annual mean visibilities in (a) spring, (b) summer, (c) autumn, and (d) winter
in Fujian Province, China. Data are from 1980–2009. Least squares regression lines at
the 95% confidence level are shown.





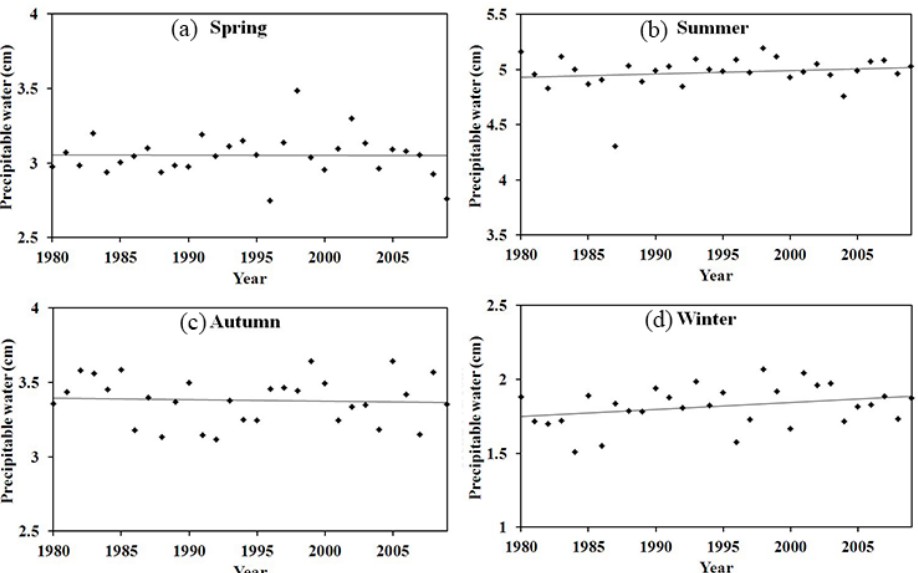


**Fig. 10.** Same as Fig. 9, except for precipitable water vapor.




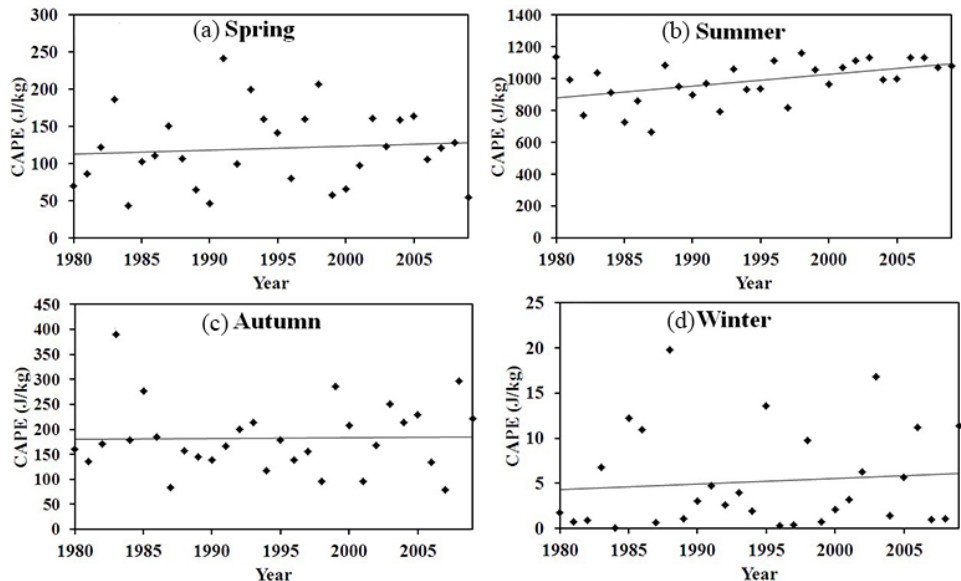


**Fig. 11.** Same as Fig. 9, except for CAPE.




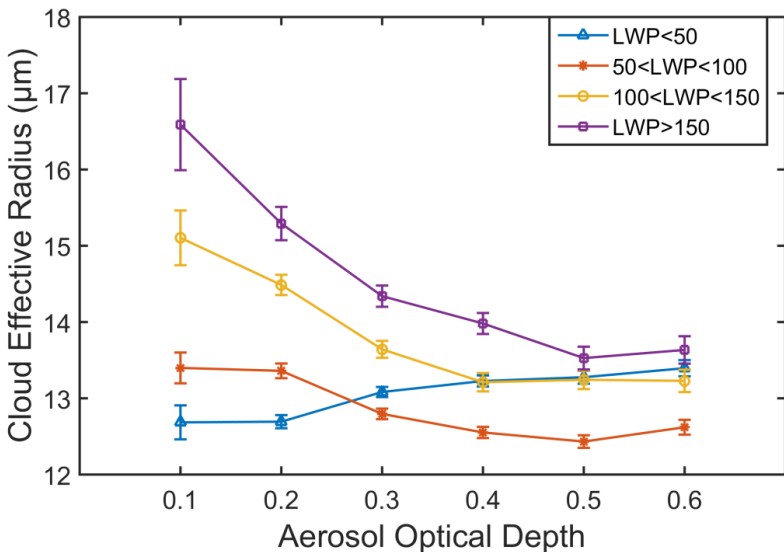


**Fig. 12.** Cloud effective radius as a function of AOD for liquid clouds (clouds with top temperatures greater than 273 K) in Fujian Province, China. Blue triangles represent cases where the LWP is less than 50 g m$^{-2}$, orange stars represent LWPs between 50 g m$^{-2}$ and 100 g m$^{-2}$, yellow circles represent LWPs between 100 g m$^{-2}$ and 150 g m$^{-2}$, and purple squares represent LWPs greater than 150 g m$^{-2}$. Error bars represent one standard error. Data are from 2003–2012.





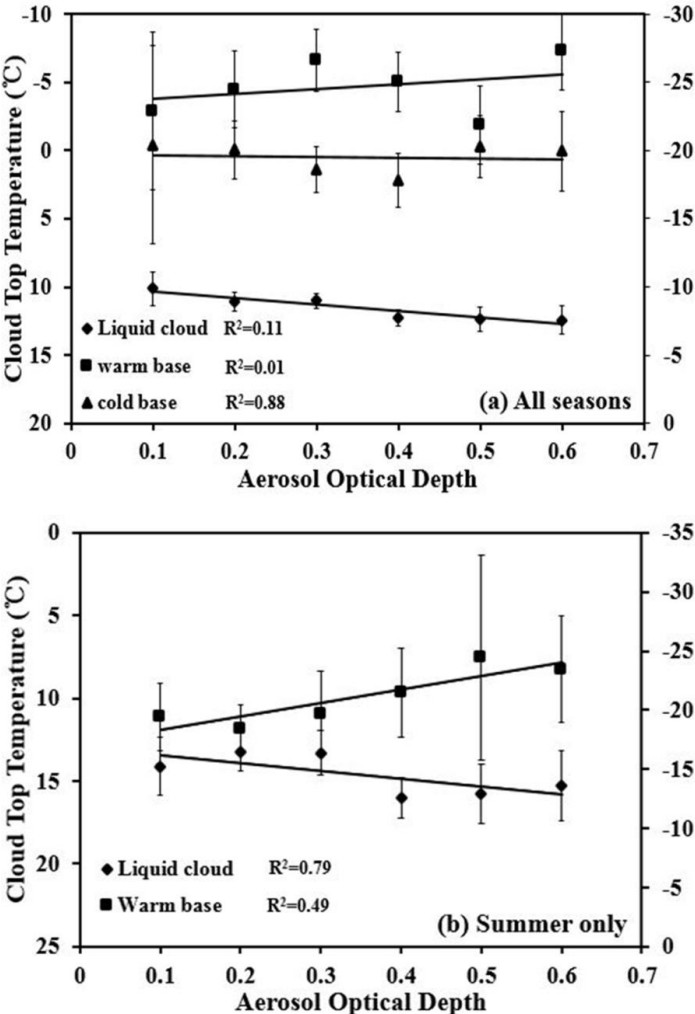

**Fig. 13.** Cloud-top temperature as a function of AOD for (a) liquid, warm-base
mixed-phase, and cold-base mixed-phase clouds in all seasons, and (b) liquid and
warm-base mixed-phase clouds in summer in Fujian Province, China. Diamonds
represent liquid clouds, squares represent warm-base mixed-phase clouds, and
triangles represent cold-base mixed-phase clouds. Right-hand ordinates are for
warm-base and cold-base mixed-phase clouds. Data are from 2006–2010.



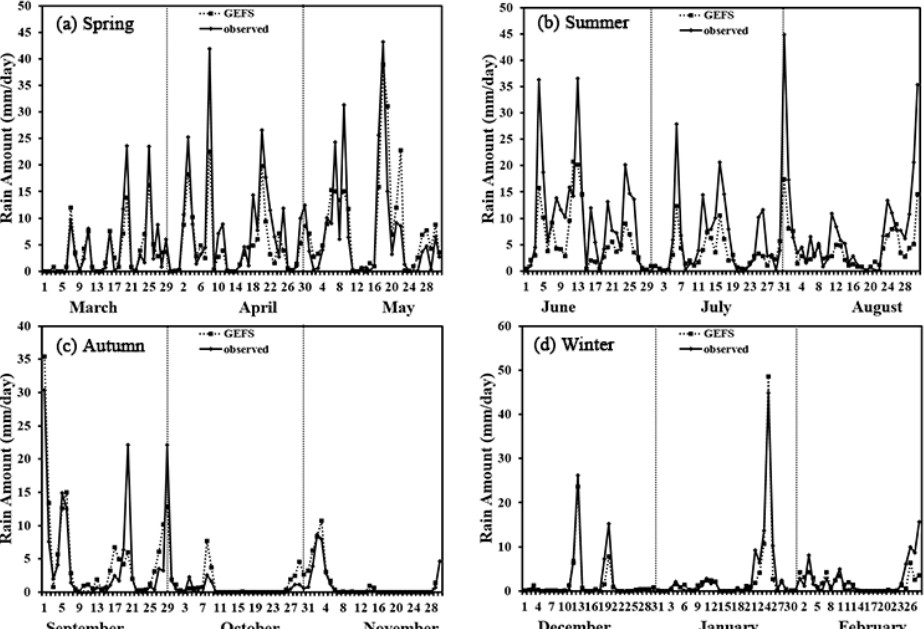

**Fig. 14.** Time series of regionally-averaged daily rainfall amount in Fujian Province,
China in (a) spring, (b) summer, (c) autumn, and (d) winter. Dotted lines represent
rainfall forecasts from the GEFS and solid lines represent rainfall measurements from
gauge-based observations. Data are from 2001.