# Peer review of "Potential Influences of Neglecting Aerosol Effects on the NCEP GFS Precipitation Forecast"

_Atmospheric Chemistry and Physics, 2017_

## Referee Comment (RC1) · Anonymous Referee #1 · 3 Jun 2017

Review of "Potential Influences of Neglecting Aerosol Effects on the NCEP GFS Precipitation Forecast" by Jiang et al. submitted for a publication in ACP

This study evaluated the potential impact of neglecting ACI on the operational rainfall forecast using ground-based and satellite observations, and NCEP GFS simulations. The main conclusion is that the ACI, which is not accounted by the forecast model, may contribute to the overestimation of light rain and underestimation of heavier rain. Since the forecast is the worst in China, the authors choose a place in China to conduct more insightful investigation using a suite of variables from gauge-based observations of precipitation, visibility, water vapor, convective available potential energy (CAPE), and satellite datasets. This is the first study to look at the potential contribution of ACI to forecast problems. The idea is new and interesting. In addition, the analysis is comprehensive. The paper is well-written and I enjoyed reading it. It is definitely worth publishing such a high-quality paper for ACP. My comments are minor generally since they would not impact the conclusions of the paper.

Major comments,

1.  About using cloud mixing ratio at 850 hpa for indicating different large-scale conditions, first, cloud water mixing ratio at such a low level would be close to zero except for boundary layer clouds (even it is not, it would not be representative of any clouds with a cloud base above 850 hpa. So, it could be problematic to use this quantify at 850 hpa. A better quantity for indicating different large-scale conditions is LWP, which can be easily obtained for both observations and model, and is typically used in many literature study.

2.  Page 23 and Figure 13, the decrease of cloud top temperature does not necessarily mean the convective invigoration as suggested by Rosenfeld et al. 2018 and then the precipitation enhancement. This is illustrated in Fan et al. 2013 (PNAS). If the CTT analyzed is for convective core only (i.e., excluding stratiform/anvil areas), this analysis may be ok. Otherwise, you can not use the increase of CTT as a proxy of convective invigoration.

3.  Discuss the data uncertainty and the implication to your results, such as satellite-retrieved AOD, the proxy of aerosols with visibility, and the rain gauge rain data. Particularly rain gauge data, it can not measure light rain with smaller rain rate such as less than 0.25 mm/h, which might contribute to the model overestimation of the light rain. Also, rain gauges might miss heavy rain spots and usually underestimate very heavy rain rate.

4.  Discuss the sampling size or sampling strategy differences between model and simulations for your analysis and the implications to your results. The observations and model data could differ in time frequency, spatial resolution, and many other things.

5.  MERRA aerosol data are not coupled with GFS simulations, discuss this caveat in the model analysis.

Specific comments,

Ln 75-79, ARI can increase precipitation at the download of the polluted places as shown in many studies (such as Carrió et al., 2010, Atmos. Res., 96, 560–574; Fan et al. 2015, GRL, 42)

Ln 95-95, I am not clear about "ARI are only considered offline and are not coupled with the dynamic system", is the temperature change by ARI considered in physics? You mentioned that aerosols are considered in the radiation scheme, which means ARI should impact radiation and temperature, and then impact dynamics. Why do you say it is not coupled with the dynamic system?

Ln 144-145, what are the major aerosol components that are chosen for both longwave and shortwave
radiative transfer calculations? It is not enough to say "one or two components".

Ln 183-184, what is the time frequency of the sounding data? If it is standard 00/12 UTC, it might not be useful.

Ln189-192, this sentence does not seem to be important unless you are specific about what new data types are included and how important they are to your analysis.

Ln229-230, 850 hPa is pretty close to the surface. Cloud mixing ratio would not exist except for boundary clouds. Do you mean total condensate mixing ratio?

Ln372-374, this is probably only true for summer time when convective clouds are dominant.

Ln 382, contradicting with a previous statement saying that AOD>0.6 is excluded from the analysis.

Page 19 and Figure 6: First, the text and Figure should be clarified about the threshold. The unit is a rain rate in text but it is a rain amount in Figure. Second, do you mean for (a) and (b), you only analyzed the data below 5 mm/hr while for (c) and (d), the data analyzed with a rate less than 20 mm/hr?  Third, the ranges of low, middle, high, and very high AOD and those of low, middle, and high cloud mixing ratios should be given. Also, needs justification why only the results in U.S. are shown.   Lastly, I do not understand why cloud mixing ratio is used. As mentioned above, cloud mixing ratio at 850 hpa does not mean much. A better quantity for indicating different conditions is LWP, which can be easily obtained from both observations and model.

Figure 12: Need to explain why cloud effective radius increases as AOD increases for LWP < 50.

Page 23 and Figure 13, the decrease of cloud top temperature does not necessarily mean the convective invigoration as suggested by Rosenfeld et al. 2018 and then the precipitation enhancement. This is illustrated in Fan et al. 2013 (PNAS). If the CTT analyzed is for convective core only (i.e., excluding stratiform/anvil areas), this analysis may be ok. Otherwise, you can not

use the increase of CTT as a proxy of convective invigoration. In addition, does the AOD used here are pre-convection value?

Line 495-497, I think this effect may only be true for summer and under the conditions that ARE is not dominant.

---

## Referee Comment (RC2) · Anonymous Referee #2 · 16 Aug 2017

This study evaluated the National Centers for Environmental Prediction (NCEP) Global Forecast System (GFS) forecast bias in different precipitation (light rain, moderate rain, heavy rain and very heavy rain) by comparing the ground-based observations in three countries. Then the correlations between GFS precipitation forecast errors and the aerosol loading are investigated extensively to examine the potential impact of neglecting aerosol-cloud-interaction (ACI) on the operational rainfall forecast. The main result is that the GFS overestimates light rain, and underestimates moderate rain, heavy rain, and very heavy rain, which is partly due to neglecting the ACI process in the GFS. The study fits within the scope of the journal, and the information and arguments are generally clear enough to be followed. Although the current study does not fully establish the causal relationship between the ACI and the bias of precipitation forecast of GFS

due partially to a lack of sufficient information, it should still be commended for confronting a highly-challenging task to make this first attempt to evaluate the numerical weather prediction forecast errors in terms of the potential effects of aerosols. Therefore, I'd recommend accepting this manuscript if the following comments are properly addressed. Major Comments: As shown in figure 3, the magnitude of underestimation in light rain and overestimation in heavy rain by GFS are all similar over three counties, but the aerosol loading in China is significantly higher than in other two countries. If the aerosol is one of the major factors causing the bias in the GFS precipitation simulation, why there is no obvious difference in the magnitudes of the bias among the three countries? For the study of the aerosol invigoration effect on the warm and cold based mixed clouds, please clarify the cloud top temperature is for convective core area or for whole convective clouds (including anvil areas). As those studies by Rosenfeld et al. [2008] and Fan et al. [2013], only the decrease of cloud top temperature for convective core with increasing of aerosol loading can be attributed to the aerosol invigoration effect. Some of descriptions are too detailed and may not be necessary. Minor Comments: Line 95: The description of "ARI are only considered offline and are not coupled with the dynamic system" is confused. Part 2.1: Since this study only used the simulation results and the details of GFS has been widely described, thus I'd suggest cutting the description in section 2.1 and paying more attention to the potential error of GFS precipitation forecast. Section 2.2.1: Such a detailed description on MERRRA-2 aerosol reanalysis is not necessary. What is the spatial resolution? Same with the CPC data? Line 251-255: Please give the observed time of the sounding data. Section 3.1.1: From figure 2, the systematic bias is found in three counties, such as the overestimations are found in north, west of China, and underestimations are found in east China. Could you explain this? Line 340: Clarify the meaning of Z. Line 385: in figure 6, please clarify the definition of the low, middle and high cloud mixing ratio, and the definition of the low, middle, high and very high AOD conditions. And why the thresholds of 5 and 20 are selected. Line 394-396: how to draw the conclusion of "the underestimation for heavy rainfall increases as AOS increases for low and middle

cloud mixing ratio conditions" from figure 6d. Line 457: Although the long-term data are used, the seasonal variations in aerosol loading, cloud properties and meteorological parameters may result in the nominal relationship as shown in figure 12. Line 479-485 and figure 13: Is the relationship statistical significant? Please give P values in figure 13. Line 485: It is either significant or not significant, based on the confidence level the authors choose. Therefore, I advise the authors to use stronger or weaker correlations, or higher or lower slopes, but not the more or less significant. Figure 8a: change the "Total" to "All"

---

## Author Comment (AC1) · 29 Aug 2017

**Review of "Potential Influences of Neglecting Aerosol Effects on the NCEP GFS Precipitation Forecast" by Jiang et al. submitted for a publication in ACP**

**This study evaluated the potential impact of neglecting ACI on the operational rainfall forecast using ground-based and satellite observations, and NCEP GFS simulations. The main conclusion is that the ACI, which is not accounted by the forecast model, may contribute to the overestimation of light rain and underestimation of heavier rain. Since the forecast is the worst in China, the authors choose a place in China to conduct more insightful investigation using a suite of variables from gauge-based observations of precipitation, visibility, water vapor, convective available potential energy (CAPE), and satellite datasets. This is the first study to look at the potential contribution of ACI to forecast problems. The idea is new and interesting. In addition, the analysis is comprehensive. The paper is well-written and I enjoyed reading it. It is definitely worth publishing such a high-quality paper for ACP. My comments are minor generally since they would not impact the conclusions of the paper.**

Thank you very much for your constructive comments and suggestions. Our point-by-point replies are given below and the corresponding revisions are shown in the revised manuscript.

**Major comments:**

1. About using cloud mixing ratio at 850 hpa for indicating different large-scale conditions, first, cloud water mixing ratio at such a low level would be close to zero except for boundary layer clouds (even it is not, it would not be representative of any clouds with a cloud base above 850 hpa. So, it could be problematic to use this quantify at 850 hpa. A better quantity for indicating different large-scale conditions is LWP, which can be easily obtained for both observations and model, and is typically used in much literature study.

*Response:*

The reason why we used the cloud mixing ratio at 850 hPa is that we focused on humidity conditions at low levels in the atmosphere. This particular level was chosen in consultation with staff at the weather stations in China. We have also used relative humidity (RH) at 850 hPa to denote large-scale humidity conditions.

We agree that LWP is a better indicator of large-scale moisture conditions, but the GFS model does not output LWP. So we calculated LWP following Yoo et al. (2012):

$$LWP = q * \rho * \Delta z,$$

where $q$ represents the cloud mixing ratio, $\rho$ represents the density of air, and $\Delta z$ is the geopotential height thickness. Only the most recent data are archived by NOAA (https://nomads.ncdc.noaa.gov/data/gfs4/). The earliest available data starts on 1 August 2015. We have downloaded one month of data and calculated the LWP.

New equitable threat and bias scores (ETS and BIAS, respectively) for the three countries were calculated under different LWP and RH scenarios. For a fixed range of LWP or RH, we further differentiate environmental conditions by choosing the top and bottom one-third of aerosol optical depth (AOD) values. The results are presented in the new Fig. 6. In Figs. 6a and 6b, ETS increase as the LWP or RH increases. This is because large-scale precipitation is diagnosed from cloud mixing ratios. ETS are smaller under polluted conditions than under clean conditions, especially when LWP or RH is high. In Figs. 6c and 6d, BIAS decrease for the polluted scenario compared with the clean scenario. The decreases in ETS and BIAS under polluted conditions suggest that AOD influences the model rainfall forecast.

[Figure]

**Fig. 6.** Equitable threat scores (a, b) and bias (BIAS) scores (c, d) as a function of precipitation amount for fixed ranges of liquid water path (LWP; a, c) and relative humidity (RH; b, d) under clean and polluted conditions. The LWP is divided into two

categories: 10–70 g m$^{-2}$ (light blue) and 70–150 g m$^{-2}$ (dark blue). Data are from August 2015 in the U.S, China, and Australia. The RH is divided into two categories: 50–70% (light green) and 70–100% (dark green). Data are from year 2015. For a given LWP or RH condition, the top and bottom one-third of AOD values are defined as polluted and clean subsets of data, respectively. The solid lines represent the clean scenario and the dotted lines represent the polluted scenario. The horizontal red lines in (c) and (d) represent perfect scores.

The following text changes were made:
Lines 184 to 187: The relative humidity (RH) at 850 hPa and the liquid water path (LWP) calculated following Yoo et al. (2012) are used, corresponding to the precipitation record in the three countries at a 0.5°x0.5° latitude-longitude resolution.

Lines 283 to 284: Under limited ranges of LWP or RH, the top and bottom one-third of AOD values denote polluted and clean subsets of data.

Lines 366 to 375: The ETS and BIAS are used to examine the model performance under clean and polluted conditions for different AOD bins with fixed LWP (Figs. 6a and 6c) or RH (Figs. 6b and 6d) in the three countries. For a particular LWP or RH condition, the top and bottom one-third of AOD values are defined as polluted and clean subsets of data. In Figs. 6a and 6b, ETS increases as the LWP or RH increases. This is because large-scale precipitation is diagnosed from cloud mixing ratios. The ETS are smaller for the polluted scenario than for the clean scenario, especially under high LWP or high RH conditions. In Figs. 6c and 6d, the BIAS decreases under polluted conditions compared with the BIAS under clean conditions. The decreases in ETS and BIAS under polluted conditions suggest that AOD influences the model rainfall forecast.

Lines 508 to 509: Equitable threat scores and BIAS scores decrease for polluted conditions.

2. Page 23 and Figure 13, the decrease of cloud top temperature does not necessarily mean the convective invigoration as suggested by Rosenfeld et al. 2008 and then the precipitation enhancement. This is illustrated in Fan et al. 2013 (PNAS). If the CTT analyzed is for convective core only (i.e., excluding stratiform/anvil areas), this analysis may be ok. Otherwise, you cannot use the increase of CTT as a proxy of convective invigoration.

*Response:*
The cloud-top temperature (CTT) obtained from CloudSat data are used to study the impact of aerosols on the cloud development of different cloud types. Based on the definition of deep mixed-phase clouds with warm bases shown in Table 1 (cloud-base temperature > 15°C), the CTT analyzed is mainly associated with the

convective core although the stratiform/anvil areas cannot be totally ignored. Both the aerosol thermodynamic effect (i.e., convective invigoration) illustrated by Rosenfeld et al. (2008) and the microphysical effect (mainly the role of more but smaller longer-lasting ice particles) emphasized by Fan et al. (2013) contribute to the decrease in CTT. The point of analyzing CTT as a function of AOD for different cloud types here is not to figure out which role is more dominant, but to find out whether the CTT decreased or increased and whether the cloud is more suitable for precipitation or not.

**Table 1.** Definitions of warm- and cold-base mixed-phase clouds and liquid clouds.

|  | Cloud-base temperature ($^{o}$C) | Cloud-top temperature ($^{o}$C) |
|---|---|---|
| Deep mixed-phase clouds with warm bases | > 15 | < -4 |
| Shallow mixed-phase clouds with cold bases | 0–15 | < -4 |
| Liquid clouds | > 0 | > 0 |

3. Discuss the data uncertainty and the implication to your results, such as satellite retrieved AOD, the proxy of aerosols with visibility, and the rain gauge rain data. Particularly rain gauge data, it cannot measure light rain with smaller rain rate such as less than 0.25 mm/h, which might contribute to the model overestimation of the light rain. Also, rain gauges might miss heavy rain spots and usually underestimate very heavy rain rate.

*Response:*

The following discussion on data uncertainties have been added to the revised manuscript:

Lines 224 to 229: Errors in satellite retrievals of AOD such as cloud contamination (Kaufman et al., 2005; Zhang et al., 2005) introduce uncertainties in the aerosol-cloud relationship (Gryspeerdt et al., 2014a, b). We use MODIS Level 3 AOD with AOD > 0.6 excluded and not the higher resolution Level 2 product to reduce the possibility of cloud contamination (Niu and Li, 2012) in AOD retrievals.

Lines 195 to 205: Visibility has been used as proxy for aerosol loading in China in several studies (Rosenfeld et al., 2007; Yang et al., 2013; Yang & Li, 2014). The main advantage is the long measurement record under all sky conditions. However, there are some limitations, e.g., the uncertainty due to humans making the observations and the influence of aerosol hygroscopic growth. To remove the humidity influence on visibility, visibility was corrected for RH (Charlson, 1969; Appel et al., 1985) using the formula adopted by Rosenfeld et al. (2007) when RH falls between 40% and 99%:

$$\frac{V_{ori}}{V_{cor}} = 0.26 + 0.4285 \, lg(100 - RH), \qquad (1)$$

where $RH$ is in percent, and $V_{ori}$ and $V_{cor}$ are the originally uncorrected and corrected visibilities, respectively. Only non-rainy data were used.

Lines 256 to 261: Rain gauge data are usually used as reference data in weather forecast and model evaluations because they come from direct physical records (Tapiador et al., 2012). The most commonly-used rain detector is the tipping bucket. Once the bucket is filled (0.1 mm), the bucket is emptied and produces a signal. This process repeats until precipitation stops. Light rain less than 0.1 mm cannot be measured. Therefore, the definition of light rain is 0.1–9.9 mm d⁻¹.

4. Discuss the sampling size or sampling strategy differences between model and simulations for your analysis and the implications to your results. The observations and model data could differ in time frequency, spatial resolution, and many other things.

*Response:*

A new 2.3.1 section entitled "Spatial and Temporal Matching of Model and Observation Data" has been added.

Lines 234 to 251: CPC-unified gauge-based daily precipitation data at a $0.5^o$ x $0.5^o$ latitude-longitude resolution in the three countries for the year 2015 are used. GFS model grid 004 data at the same latitude-longitude resolution ($0.5^o$ x $0.5^o$) are also used. Forecast precipitation for a one-day accumulation generated at three-hourly intervals (e.g., at 03, 06, 09, 12, 15, 18, 21, 24 UTC), starting from the control time of 00 UTC, are used to match the corresponding gauge-based observations. The MERRA-2 aerosol analysis is not coupled with GFS simulations. Daily MERRA-2 AOD is at a resolution of $0.625^o$ x $0.5^o$ and is interpolated to the CPC and GFS precipitation resolution using a linear interpolation method. The spatial and temporal resolutions of the matched data sets are $0.5^o$ x $0.5^o$ and are generated for each day. There are ~3 686 000 grid points in total.

For the long-term analysis focused on Fujian, China, the NWP model reforecast precipitation amount accumulated over the period of 12 hours to 36 hours out from the 00 UTC run at six-hourly intervals and at a $1^o$ x $1^o$ latitude-longitude resolution for the years 1985 to 2010 are used to calculate the modeled daily precipitation amount in each grid box. They are interpolated to match the long-term ground-based precipitation observations recorded at each of the 67 stations in the study region of Fujian, China (Fig. 1). There are 9495 days in total with matched data.

5. MERRA aerosol data are not coupled with GFS simulations. Discuss this caveat in

the model analysis.

*Response:*

This statement has been added.

Lines 239 to 240: The MERRA-2 aerosol analysis is not coupled with GFS simulations.

**Specific comments:**

1.Ln 75-79, ARI can increase precipitation at the download of the polluted places as shown in many studies (such as Carrió et al., 2010, Atmos. Res., 96, 560–574; Fan et al. 2015, GRL, 42)

*Response:*

This statement has been added.

Lines 78 to 79: The suppressed convection by ARI may also lead to rainfall enhancement downwind of polluted places (Carrió et al., 2010; Fan et al., 2015).

2. Ln 95-95, I am not clear about "ARI are only considered offline and are not coupled with the dynamic system", is the temperature change by ARI considered in physics? You mentioned that aerosols are considered in the radiation scheme, which means ARI should impact radiation and temperature, and then impact dynamics. Why do you say it is not coupled with the dynamic system?

*Response:*

A seasonal climatological tropospheric aerosol background with a large horizontal resolution is used for both longwave and shortwave radiation. There is a current effort underway to change this to a monthly background. The temperature change caused by aerosols is not coupled to each forecast interval. Therefore, it is not coupled with the dynamic system.

3. Ln 144-145, what are the major aerosol components that are chosen for both longwave and shortwave radiative transfer calculations? It is not enough to say "one or two components".

*Response:*

There are five species considered in the radiative transfer calculation, namely, dust, sea salt, sulfates, organic carbon, and black carbon, which are similar to those in the GOCART model. A generalized map of various aerosol components was

constructed, and then in each grid, one or two major components (based on climatology) were chosen to compute radiative properties in each of the radiation spectral bands.

Lines 129 to 130: as the sentence was revised as follows: "One or two major components in each grid (based on climatology) were chosen for both longwave and shortwave radiative transfer calculations."

4. Ln 183-184, what is the time frequency of the sounding data? If it is standard 00/12 UTC, it might not be useful.

***Response:***

It is the standard 00/12 UTC set of soundings and the only available sounding data we have to use.

5. Ln189-192, this sentence does not seem to be important unless you are specific about what new data types are included and how important they are to your analysis.

***Response:***

The sentence has been deleted. Also, we have also followed another reviewer's suggestion to shorten the description of the MERRA-2.

6. Ln229-230, 850 hPa is pretty close to the surface. Cloud mixing ratio would not exist except for boundary clouds. Do you mean total condensate mixing ratio?

***Response:***

We have used LWP and RH for better representing large-scale conditions. Please see the response to Major Comment 1 for more details.

7. Ln372-374, this is probably only true for summer time when convective clouds are dominant.

***Response:***

It is true that the heavy rain enhancement is mostly seen in summer when convective clouds are dominant. In the specific analysis of the correlation coefficients of visibility and rain amount (Table 3) and rain frequency (Table 4) in Fujian Province, China, the aerosol effect on heavy rain enhancement is significant in summertime.

8. Ln 382, contradicting with a previous statement saying that AOD>0.6 is excluded

from the analysis.

*Response:*

Two AOD datasets are used in the study. One dataset is the MERRA-2 Aerosol Reanalysis, which is used in the three-country analysis and where AOD > 0.6 are not excluded. The other dataset is the MODIS Level 3 AOD product, which is used in the Fujian analysis. Satellite-retrieved AOD > 0.6 are excluded in that analysis to reduce the possibility of cloud contamination in the AOD retrievals.

9. Page 19 and Figure 6: First, the text and Figure should be clarified about the threshold. The unit is a rain rate in text but it is a rain amount in Figure. Second, do you mean for (a) and (b), you only analyzed the data below 5 mm/hr while for (c) and (d), the data analyzed with a rate less than 20 mm/hr? Third, the ranges of low, middle, high, and very high AOD and those of low, middle, and high cloud mixing ratios should be given. Also, needs justification why only the results in U.S. are shown. Lastly, I do not understand why cloud mixing ratio is used. As mentioned above, cloud mixing ratio at 850 hpa does not mean much. A better quantity for indicating different conditions is LWP, which can be easily obtained from both observations and model.

*Response:*

Figure 6 have been revised. First, the units stated in the text and in the figure are now the same. Second, a threshold is used in the contingency table when calculating ETS and BIAS. The definition of hits or misses is based on the forecast rain amount above a certain threshold. In the new Figure 6, more thresholds are used. Third, the cloud mixing ratio at 850 hPa is replaced by LWP and RH in the new Figure 6. For certain LWP or RH conditions, the top and bottom one-third of AOD values are defined as polluted and clean subsets of data. Also, results for three countries are now shown.

10. Figure 12: Need to explain why cloud effective radius increases as AOD increases for LWP < 50.

*Response:*

Figure 12: Clouds with LWP < 50 m$^{-2}$ are not thick. The MODIS sensor may have larger uncertainties when dealing with thin clouds. Also, when LWP < 50 m$^{-2}$, the ambient saturation may not exceed the critical saturation, so cloud droplets are not yet activated. The cloud effective radius may then increase as AOD increases. Stratus clouds may be more influenced by environmental thermodynamic or other factors.

11. Page 23 and Figure 13, the decrease of cloud top temperature does not necessarily mean the convective invigoration as suggested by Rosenfeld et al. 2018 and then the precipitation enhancement. This is illustrated in Fan et al. 2013 (PNAS). If the CTT analyzed is for convective core only (i.e., excluding stratiform/anvil areas), this analysis may be ok. Otherwise, you cannot use the increase of CTT as a proxy of convective invigoration. In addition, does the AOD used here are pre-convection value?

*Response:*

The cloud-top temperature (CTT) obtained from CloudSat data are used to study the impact of aerosols on the cloud development of different cloud types. Based on the definition of deep mixed-phase clouds with warm bases shown in Table 1 (cloud-base temperature $> 15^{o}$C), the CTT analyzed is mainly associated with the convective core although the stratiform/anvil areas cannot be totally ignored. Both the aerosol thermodynamic effect (i.e., convective invigoration) illustrated by Rosenfeld et al. (2008) and the microphysical effect (mainly the role of more but smaller longer-lasting ice particles) emphasized by Fan et al. (2013) contribute to the decrease in CTT. The point of analyzing CTT as a function of AOD for different cloud types here is not to figure out which role is more dominant, but to find out whether the CTT decreased or increased and whether the cloud is more suitable for precipitation or not.

**Table 1.** Definitions of warm- and cold-base mixed-phase clouds and liquid clouds.

|  | Cloud-base temperature ($^{o}$C) | Cloud-top temperature ($^{o}$C) |
| --- | --- | --- |
| Deep mixed-phase clouds with warm bases | > 15 | < -4 |
| Shallow mixed-phase clouds with cold bases | 0–15 | < -4 |
| Liquid clouds | > 0 | > 0 |

AOD data used here are daily means so it is difficult to say if this data is pre-convective or not.

12. Line 495-497, I think this effect may only be true for summer and under the conditions that ARE is not dominant.

*Response:*

Lines 472 to 473: It is true that heavy rain enhancement occurs mainly in the summer and under the condition that ARE is not dominant. In the analysis of Fig. 14, lines 479 to 481: "… modeled precipitation amounts are significantly less than observed precipitation amounts over the region in summer when deep convective clouds and heavy to very heavy rain tends to occur."

---

## Author Comment (AC2) · 29 Aug 2017

**Comments on the manuscript titled “Potential Influences of Neglecting Aerosol Effects on the NCEP GFS Precipitation Forecast” by Jiang et al.**

This study evaluated the National Centers for Environmental Prediction (NCEP) Global Forecast System (GFS) forecast bias in different precipitation (light rain, moderate rain, heavy rain and very heavy rain) by comparing the ground-based observations in three countries. Then the correlations between GFS precipitation forecast errors and the aerosol loading are investigated extensively to examine the potential impact of neglecting aerosol-cloud-interaction (ACI) on the operational rainfall forecast. The main result is that the GFS overestimates light rain, and underestimates moderate rain, heavy rain, and very heavy rain, which is partly due to the neglecting ACI process in GFS. The study fits within the scope of the journal, and the information and arguments are generally clear enough to be followed. Although the current study does not fully established the causal relationship between the ACI and the bias of precipitation forecast of GFS due partially to a lack of sufficient information, it should still be commended for confronting a highly-challenging task to make this first attempt to evaluate the numerical weather prediction forecast errors in terms of the potential effects of aerosols. Therefore, I'd recommend accepting this manuscript if the following comments are properly addressed.

Thank you very much for your constructive comments and suggestions. Our point-by-point replies are given below and the corresponding revisions are shown in the revised manuscript.

**Major Comments:**

1. As shown in figure 3, the magnitude of underestimation in light rain and overestimation in heavy rain by GFS are all similar over three counties, but the aerosol loading in China is significantly higher than in other two countries. If the aerosol is one of the major factors causing the bias in the GFS precipitation simulation, why there is no obvious difference in the magnitudes of the bias among the three countries?

***Response:***

First, the intention of Figure 3 is to show that the GFS model overestimates light rain and underestimates heavier rain. Second, of course, these model biases are caused by many factors, including initial dynamic settings and weather regimes. But it is beyond the scope of this paper to explore all possible causes. Comparing the model performance globally according to aerosol loading only is not sufficient because the model performance may differ for different regions. Our focus is on identifying any potential contribution of neglecting aerosol effects to the biases. The relationship between model performance and AOD was thus further investigated. This is also why we compared results from three countries. In each country, the standard deviation of the daily precipitation difference as a function of aerosol optical depth is presented in Fig. 5. Each

point represents a grid box. The significant positive correlation between standard deviation and AOD illustrates that neglecting aerosol effects may contribute to the model forecast bias. Third, the non-linear impact of aerosols on precipitation may also differ according to meteorological conditions, aerosol components, and the interactions between thermal and dynamic conditions. This is why we then focused on one specific region, Fujian Province, and did a long-term statistical evaluation of rainfall forecasts to mitigate these fluctuations in the model forecast accuracy.

2. For the study of the aerosol invigoration effect on the warm and cold based mixed clouds, please clarify the cloud top temperature is for convective core area or for whole convective clouds (including anvil areas). As those studies by Rosenfeld et al. [2008] and Fan et al. [2013], only the decrease of cloud top temperature for convective core with increasing of aerosol loading can be attributed to the aerosol invigoration effect.

**Response:**

The cloud-top temperature (CTT) obtained from CloudSat data are used to study the impact of aerosols on the cloud development of different cloud types. Based on the definition of deep mixed-phase clouds with warm bases shown in Table 1 (cloud-base temperature  $> 15^{\circ}\text{C}$ ), the CTT analyzed is mainly associated with the convective core although the stratiform/anvil areas cannot be totally ignored. Both the aerosol thermodynamic effect (i.e., convective invigoration) illustrated by Rosenfeld et al. (2008) and the microphysical effect (mainly the role of more but smaller longer-lasting ice particles) emphasized by Fan et al. (2013) contribute to the decrease in CTT. The point of analyzing CTT as a function of AOD for different cloud types here is not to figure out which role is more dominant, but to find out whether the CTT decreased or increased and whether the cloud is more suitable for precipitation or not.

**Table 1.** Definitions of warm- and cold-base mixed-phase clouds and liquid clouds.

|                                            | Cloud-base temperature
( $^{\circ}\text{C}$ ) | Cloud-top temperature
( $^{\circ}\text{C}$ ) |
|--------------------------------------------|--------------------------------------------------|-------------------------------------------------|
| Deep mixed-phase clouds with warm bases    | $> 15$                                           | $< -4$                                          |
| Shallow mixed-phase clouds with cold bases | $0-15$                                           | $< -4$                                          |
| Liquid clouds                              | $> 0$                                            | $> 0$                                           |

3. Some of descriptions are too detailed and may not be necessary.

**Response:**

We have modified the descriptions accordingly. A brief description of the model setting, which is relevant to this study, has been given. Also, detailed descriptions of the MERRA-2 analysis in section 2.2.1 have been shortened.

**Minor Comments:**

1. Line 95: The description of “ARI are only considered offline and are not coupled with the dynamic system” is confused.

**Response:**

A seasonal climatological tropospheric aerosol background with a large horizontal resolution is used for both longwave and shortwave radiation. There is a current effort underway to change this to a monthly background. The temperature change caused by aerosols is not coupled to each forecast interval. Therefore, it is not coupled with the dynamic system.

2. Part 2.1: Since this study only used the simulation results and the details of GFS has been widely described, thus I'd suggest cutting the description in section 2.1 and paying more attention to the potential error of GFS precipitation forecast.

**Response:**

Lines 121 to 144: We have modified the descriptions accordingly. A brief description of the model setting, which is relevant to this study, has been given.

3. Section 2.2.1: Such a detailed description on MERRA-2 aerosol reanalysis is not necessary. What is the spatial resolution? Same with the CPC data?

**Response:**

Lines 157 to 167: This part of the manuscript has been shortened. The spatial resolution of the MERRA-2 reanalysis is  $0.625^{\circ} \times 0.5^{\circ}$  and that of CPC data is  $0.5^{\circ} \times 0.5^{\circ}$ . The data matching strategy is described in the newly-added section 2.3.1.

4. Line 251-255: Please give the observed time of the sounding data.

**Response:**

It is twice a day (at 00 UTC and 12 UTC). This information has been added to line 207.

5. Section 3.1.1: From figure 2, the systematic bias is found in three counties, such as the overestimations are found in north, west of China, and underestimations are found in east China. Could you explain this?

***Response:***

The GFS model tends to overestimate light rain and underestimate heavier rain. In the northern and western parts of China, it seldom rains and when it rains, it is mainly light rain. So the GFS model tends to overestimate precipitation in these parts of China. In eastern China, it rains more and deep convective precipitation is common. So the GFS model tends to underestimate rain in this region.

6. Line 340: Clarify the meaning of Z.

***Response:***

Line 325: The Z-score is the number of standard deviations from the mean value of the reference population. When 95% of the values fall within two standard deviations from the mean, a normal probability distribution is defined (according to the 68-95-99.7 rule). The p value is set as 0.05 in this study, therefore, the mean difference is not significant at a two-sigma level when  $Z

**Fig. 6.** Equitable threat scores (a, b) and bias (BIAS) scores (c, d) as a function of precipitation amount for fixed ranges of liquid water path (LWP; a, c) and relative humidity (RH; b, d) under clean and polluted conditions. The LWP is divided into two categories:  $10\text{--}70\text{ g m}^{-2}$  (light blue) and  $70\text{--}150\text{ g m}^{-2}$  (dark blue). Data are from August 2015 in the U.S., China, and Australia. The RH is divided into two categories: 50–70% (light green) and 70–100% (dark green). Data are from year 2015. For a given LWP or RH condition, the top and bottom one-third of AOD values are defined as polluted and clean subsets of data, respectively. The solid lines represent the clean scenario and the dotted lines represent the polluted scenario. The horizontal red lines in (c) and (d) represent perfect scores.

8. Line 394-396: how to draw the conclusion of “the underestimation for heavy rainfall increases as AOD increases for low and middle cloud mixing ratio conditions” from figure 6d.

**Response:**

This sentence has been deleted.

9. Line 457: Although the long-term data are used, the seasonal variations in aerosol loading, cloud properties and meteorological parameters may result in the nominal relationship as shown in figure 12.

***Response:***

Line 434: Seasonal variations in aerosol loading, cloud properties, and meteorological parameters may influence aerosol-cloud-precipitation interactions. This is why we examine the impact of aerosols on clouds and precipitation for certain cloud types and ranges of LWP values. In Figure 12, the cloud effective radius as a function of AOD under different LWP conditions for liquid clouds is shown. The randomly-mixed samples are rearranged according to AOD. The figure shows some perturbations caused by changes in AOD.

10. Line 479-485 and figure 13: Is the relationship statistical significant? Please give P values in figure 13.

***Response:***

We have included P values in the new Figure 13.

**Fig. 13.** Cloud-top temperature as a function of aerosol optical depth for (a) liquid, warm-base mixed-phase, and cold-base mixed-phase clouds in all seasons, and (b) liquid and warm-base mixed-phase clouds in summer in Fujian Province, China. Diamonds represent liquid clouds, squares represent warm-base mixed-phase clouds, and triangles represent cold-base mixed-phase clouds. Right-hand ordinates are for warm-base and cold-base mixed-phase clouds. Data are from 2006–2010.

11. Line 485: It is either significant or not significant, based on the confidence level the authors choose. Therefore, I advise the authors to use stronger or weaker correlations, or higher or lower slopes, but not the more or less significant.

***Response:***

Lines 461 to 464: This sentence has been rewritten as “The negative slope of the linear relationship between CTT and AOD for warm-base mixed-phase clouds and the positive slope of the linear relationship between CTT and AOD for liquid clouds are both stronger in summer (Fig. 13b).”

12. Figure 8a: change the “Total” to “All”

***Response:***

Done.

---

## Author Comment (AC4) · 29 Aug 2017

The comment was uploaded in the form of a supplement:
https://www.atmos-chem-phys-discuss.net/acp-2017-256/acp-2017-256-AC4-supplement.pdf